# The Features of Distribution of Chemical Elements, including Heavy Metals and Cs-137, in Surface Sediments of the Barents, Kara, Laptev and East Siberian Seas

**Dmitry F. Budko** [1], **Liudmila L. Demina** [1,*], **Anna V. Travkina** [2], **Dina P. Starodymova** [1] **and Tatiyana N. Alekseeva** [1]

1    Shirshov Institute of Oceanology, RAS, 36 Nakhimovsky Prospekt, 117997 Moscow, Russia;
     dmitry.b-1990@yandex.ru (D.F.B.); d.smokie@gmail.com (D.P.S.); tania@blackout.ru (T.N.A.)
2    Vernadsky Institute of Geochemistry and Analytical Chemistry, RAS, 19 Kosygina St.,
     119991 Moscow, Russia; a_travkina@mail.ru
*    Correspondence: l_demina@mail.ru; Tel.: +7(963)608-17-47

**Abstract:** Over the recent few decades, due to climate warming and the continuing exploration of Arctic seas' mineral resources, the scientific interest in contamination problems has deepened significantly. In this study, for the first time, we characterize the distribution features of 47 elements (major and trace elements, including heavy metals, metalloid As, and Cs-137 technogenic radionuclide) in surface bottom sediments from some areas of the Barents, Kara, Laptev, and East-Siberian Seas. The lithogenic material was the main factor that controlled variability in many elements (Be, Al, Ti, Cr, Ga, Rb, Sr, Y, Zr, Nb, Ba, REE, Pb, Th, U, W, and Cs). Among the hydrogenic processes, the formation of Fe and Mn oxyhydroxides has the greatest impact on the Mn, Fe, Co, Ni, Cu, Ge, and Mo, and insignificantly V and Sb, variability in sediments. These, along with minor to moderate values of enrichment factor (EF) for most elements, allowed us to conclude that the observed element distribution is related to predominantly natural processes of thermal abrasion, river-run, and atmospheric input. The exception is As, which exhibited the elevated EF (up to 20) in the western and central Kara Sea, as well as in the Vilkitsky Strait. Since no significant relationship between As and Fe and Mn oxyhydroxides distribution was found, we may assume primarily an anthropogenic source of As, related to the peat and/or coal combustion. According to the criteria of Ecological Risks assessment, all the examined areas have a low degree of risk. Data on the specific activity of Cs-137 correspond to the background average values characteristic for these regions. The highest levels of Cs-137 concentration (Bq/kg) were detected in the sediments of the Ob and Yenisei Rivers' estuaries.

**Keywords:** Barents Sea; Kara Sea; Laptev Sea; East Siberian Sea; bottom sediments; radionuclides; major and trace elements; criteria of contamination; potential ecological risk





## 1. Introduction

The development of the resource potential of the Arctic seas is associated with an increase in the anthropogenic impact on the environment. Of particular concern is the increasing pollution of coastal and shelf areas by heavy metals and human-made radionuclides [1]. Natural climate change is reflected primarily in high latitudes areas. According to the ten-year monitoring of the hydrological characteristics of the Kara Sea, an annual increase in the summer deglaciation of the ice sheet was observed [2]. The reduction of the ice cover, in turn, leads to desalination of the Arctic seas, as well as a partial transformation of the chemical composition of the water. As a result, the forms of migration of potentially toxic elements, in particular heavy metals and radionuclides, may also change.

Since the early 1990s, the interest of the scientific community in the problem of radioactive contamination of the Arctic due to climate warming has increased significantly. The

Arctic is characterized by a variety of sources and ways of technogenic radionuclides pollution associated with the consequences of mass tests of nuclear weapons in the atmosphere in the past, during which the Arctic region (up to the 70° N) was contaminated with about 4.3 Pb Cs-137 [3], as well as the operation of nuclear fuel reprocessing activities at Sellafield, UK, a reprocessing plant at Cap de la Hague in France [4], Russian processing plants located in the basins of the Ob and Yenisei Rivers [5], and with the Chernobyl accident. In addition, a few burials of solid radioactive waste in the bays of the Novaya Zemlya Archipelago's eastern coast might be a local source of pollution. Due to the great danger of technogenic radionuclides (in particular Cs-137) to natural ecosystems, as a consequence of their ecotoxicity, the acknowledgment of their distribution features in Arctic sediments is an urgent modern task. Despite that, data on the content of heavy metals and radioactive nuclides in the sediments of the Eurasian Arctic shelf seas remain not completely investigated to date.

First investigations of the chemical element composition of the Arctic bottom sediments were conducted in the 1990s, including the Barents Sea [6–8], Kara Sea [9–11], and Laptev Sea [12–15]. Over the past decade, the geochemical characteristics of the Kara Sea bottom sediments were examined by the X-ray fluorescent method and focused on the major elements, and in a few cases, the trace element composition [16–18]. Data on the total content of trace elements obtained by the more sensitive methods (ICP-MS, ICP-AES, AAC) are available only for some areas of the Kara Sea [19,20]. For the Barents Sea surface sediments, estimation of a regional background content of some heavy metals and their anthropogenic loads was performed [21]. In the Barents Sea surface sediments, the element distribution exhibited a noticeable dependence on grain size composition, which in turn is related to bottom topography and hydrodynamics conditions; for potentially toxic elements, such as Pb, Cu, and Zn, a rather weak anthropogenic influence was recorded, except for As, whose average content (19 ppm) exceeds the UCC values (8 ppm) [22]. The high concentrations of Cr, Zn, Ni, Cu, and As were detected in the deeper parts of the East Siberian and Laptev Seas and the estuarine shelves of the Lena and Yana Rivers, while the high Cd content was observed in sediments of the eastern East Siberian Sea [23]. These authors based on the contamination indices also found the slight As enrichment, possibly originating from natural sources and no contamination by Cr, Ni, Cu, Zn, Pb, and Cd.

The purposes of this work are as follows: in the surface sediments from different areas of the Barents, Kara, Laptev, and East Siberian Seas (i) to evaluate a current state of trace metal contamination applying contamination indices; (ii) to estimate the factors which influence trace metals' distribution using multivariate statistical analyses; (iii) to study features of the spatial distribution of Cs-137.

## 2. Materials and Methods

### 2.1. Some Geochemical Features of the Studied Areas

In this study, we examined the total content of elements including radioactive isotope Cs-137, in sediments of the shelf zone of the Barents, Kara, Laptev, and East Siberian Seas. Each of the studied marine areas has its natural features and anthropogenic influence, which is of interest in a comparative study of the geochemical properties of bottom sediments. The Barents Sea is a shallow part of the North European Arctic shelf, which is heavily influenced by the Atlantic Ocean. Nutrient input from the relatively warm and salty Atlantic waters results in primary production development in this part of the Arctic. According to some estimates, about 40% of the organic matter of the entire Arctic shelf is produced in the Barents Sea [24]. At the same time, the strengthening of the "atlantification" of the Barents Sea, observed over the past decade, is accompanied by a reduction in the ice cover, and a general restructuring of the hydrological, chemical, and biological structures of the Barents Sea ecosystems [25,26]. With an insignificant river runoff, the atmospheric transport, ice discharge, and advection of water masses from Western Europe serve as the major sources of chemical elements' input into the Barents Sea [21,27].

Unlike the Barents Sea, the Siberian Arctic Seas (Kara, Laptev, and East Siberian) are influenced by the massive freshwater runoff. Only in the Kara Sea, the river runoff of the

Ob and Yenisei Rivers supplies about 45% of all freshwater (1290 km$^3$/year), discharged into the Arctic Ocean, which greatly affects the thermohaline stratification of the water column of this sea basin [2]. Massive river runoff, as well as the radioactive waste burial grounds in the northwestern part of the sea and industrial development of hydrocarbon resources, determines the main environmental risks of the Kara Sea [28]. As we move eastward, the role of the ice sedimentation in the redistribution and migration of suspended material from coastal areas into the interior of the ocean increases [29]. The Laptev Sea is considered as the main source of pack ice which transports suspended particles' load through Transpolar Drift in the Arctic Ocean [30]. The Lena River runoff and the smaller rivers (in total 720 km$^3$/year), which makes up about a quarter of the total freshwater inflow into the Arctic Ocean, plays an important role in the formation of natural character of the Laptev Sea and the inflow of chemical elements [31].

The East Siberian Sea is characterized by the longest ice-covered period among the Russian Arctic Seas. The river runoff into the East Siberian Sea is negligible, amounting to 230–240 km$^3$/year, or about 9% of all freshwater runoff into the Arctic Ocean [31]. However, it is the river runoff that can determine the main environmental risks of the East Siberian Sea's basin. The mining enterprises (non-ferrous metal ores and coal) located on the catchment area and coasts of the Kolyma and Indigirka Rivers serve a source of Cu, Fe, Zn, and some other heavy metals transported by the rivers into the East Siberian Sea [32]. Coastal abrasion processes also play an important role in the supply of chemical elements into the Arctic seas [33].

### 2.2. Sampling Location and Processing

For the chemical elements' analysis, 24 samples of surface bottom sediments (horizon of 0–2 cm) were taken with a box-corer in cruises of R/V "Akademik Mstislav Keldysh" (2016, 2017 and 2019) in different parts of the Barents, Kara, Laptev, and East Siberian seas (Figure 1). The coordinates and depth of sampling stations are listed in Table 1.

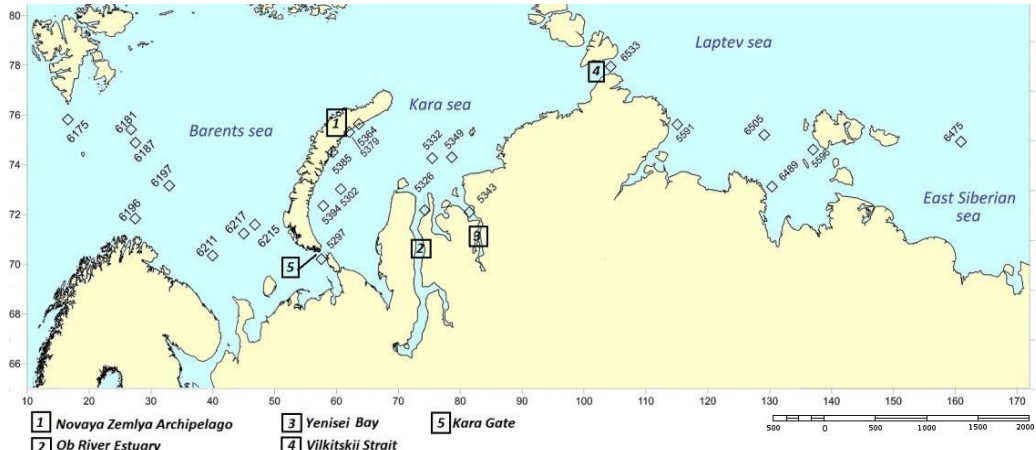

**Figure 1.** The sampling stations location of surface bottom sediments from the Arctic seas.

Stations 5326, 5332, 5343, and 5349 were located at shallow depths (13–34 m) in the area of marginal filters of the Ob and Yenisei Rivers, where sedimentation of the river runoff suspended material predominates. Stations 6475 and 6505 are located at intermediate depths in the open parts of the East Siberian and Laptev Seas, respectively. The deeper stations (124–330 m) were located in the Kara Gate (St. 5297) and Vilkitsky (St. 6533) Straits, as well as in the area of the Novaya Zemlya Trough in the Kara Sea, where sedimentary material is commonly accumulated (St. 5394). In the Barents Sea, only one station 6215 is located in the shallow part of the sea (the Goose Bank), while the others were sampled in the seabed depressions (depth of >150 m).

Onboard the ship, the collected samples were placed into the plastic bags and stored in a frozen state (−18 °C).

**Table 1.** Location of the surface bottom sediments sampling (horizon of 0–2 cm).

| Cruise's Number and Year | Station | Latitude, N | Longitude, E | Area | Depth, m |
|---|---|---|---|---|---|
| No 66, 2016 | 5297 | 70°11.7′ | 057°32.9′ | Kara Gate Strait | 124 |
| | 5302 | 73°06.1′ | 061°19.1′ | Kara Sea | 87 |
| | 5326 | 72°09.9′ | 074°17.7′ | Ob River Estuary | 13 |
| | 5332 | 74°15.9′ | 075°29.2′ | Kara Sea, central part | 30 |
| | 5343 | 72°05.6′ | 081°28.9′ | Yenisei Bay | 11 |
| | 5349 | 74°17.9′ | 078°38.0′ | Kara Sea, central part | 34 |
| | 5364 | 75°38.3′ | 063°37.4′ | Novaya Zemlya, Blagopoluchiya Bay | 85 |
| | 5379 | 74°38.5′ | 059°53.7′ | Novaya Zemlya, Sedov Bay | 51 |
| | 5385 | 74°30.8′ | 059°24.0′ | Novaya Zemlya, Oga Bay | 100 |
| | 5394 | 72°20.8′ | 057°52.7′ | Novaya Zemlya Trough | 330 |
| No 69, 2017 | 5591 | 75°41.2′ | 115°45.5′ | Laptev Sea | 45 |
| | 5596 | 74°25.0′ | 130°49.8′ | Laptev Sea | 22 |
| No 75, 2019 | 6175 | 75°50.451′ | 16°35.624′ | Barents Sea | 360 |
| | 6181 | 75°26.157′ | 26°48.459′ | Barents Sea | 199 |
| | 6187 | 74°53.936′ | 27°31.685′ | Barents Sea | 351 |
| | 6196 | 71°49.95′ | 27°30.111′ | Barents Sea | 329 |
| | 6197 | 73°9.899′ | 32°59.965′ | Barents Sea | 209 |
| | 6211 | 70°19.979′ | 40°0.222′ | Barents Sea | 168 |
| | 6215 | 71°35.219′ | 46°50.598′ | Barents Sea | 53 |
| | 6217 | 71°13.527′ | 45°2.444′ | Barents Sea | 240 |
| No 78, 2019 | 6475 | 74°55.4′ | 160°56.5′ | East Siberian Sea | 45 |
| | 6489 | 73°07.1′ | 130°23.8′ | Lena River delta | 24 |
| | 6505 | 75°12.0′ | 129°08.6′ | Laptev Sea | 41 |
| | 6533 | 77°57.1′ | 104°19.2′ | Vilkitsky Strait | 228 |

*2.3. Analytical Methods*

For most stations, a comprehensive examination of elements was carried out, while only carbon content determination was performed at two stations in the Kara Sea (5302 and 5379) and two stations in the Laptev Sea (5591 and 5596).

2.3.1. Grain-Size and Elemental Analysis of Sediment Samples

In the stationary laboratory, the grain size analysis in sediments of natural wetness was carried out. The grain-size composition was determined by the classic method of the water-mechanical analysis [34]. Before chemical treatment, samples of sediments were dried at a temperature of 55 °C in the oven, followed by grounding in an agate mortar.

To determine the total content of chemical elements, a complete acid decomposition of samples in PFA vials in Microwave System "Speed Wave" was used (Berghof Products, Eningen, Germany) according to [35]. About 100 mg of the dry powdered sediment was placed in PFA vials of 4 mL volume (Savillex, Eden Prairie, MN, USA), where 1.5 mL of a mixture of concentrated acids HF: $HNO_3$ in a ratio of 5: 1 and 0.5 mL of HCl were added. The ultra-pure nitric acid "Premium-Grade" (Merck, Kenilworth, NJ, USA) specialized for analytical work was used, while fluoric and hydrochloric acids were purified through the sub-boiling distillation system Berghof BSB-939-IR (Berghof Products, Eningen, Germany). To avoid artificial contamination while handling and analysis of trace elements, all the flasks and laboratory glassware were pre-cleaned with 10% $HNO_3$ solution. For every

7 specimens, one blank analysis was performed. A decomposition of the examined samples, as well as the certified reference material, and blank samples, was carried out in the microwave system MWS Speed Wave (Berghof Products, Eningen, Germany) in the following mode: 5 min at a high intensity of study of 800 W and a temperature of 170 °C, followed by a lower radiation intensity and a temperature of 170 °C during 40 min. Then, the vials were placed on a Teflon heating platform and evaporated at a temperature of 70 °C three times with the addition of 1 mL of HCl to dissolve fluorides, formed during the decomposition of samples with HF. After the second evaporation, 0.5 mL of $HClO_4$ "Suprapure" (Merck, Kenilworth, NJ, USA) was added into vials to decompose soot particles. The residue was dissolved in 1 mL of HCl and brought to a final volume (25 mL) with a solution of the 3% $HNO_3$. Concentrations of all chemical elements, except Fe, were determined by inductively coupled plasma mass spectrometry (ICP-MS) on the Agilent 7500 spectrometer (Agilent Technologies, Inc., Santa Clara, CA, USA) using the internal In standard. The determination of Fe was carried out by the flame atomic absorption spectrometry on the spectrometer "KVANT-2A" (Kortek, St. Petersburg, Russia).

Quality control of the analyses was evaluated using standard reference material (SRM) of the National Institute of Standards and Technology (Gaithersburg, MD, USA) NIST 2702 (inorganics in marine sediments). Recovery and precision for all certified elements were calculated based on the three replicate measurements of the SRM sample (Table S1). A good precision within 10% was obtained for most elements except Ti, Cr, As, Mo, and Ag (10–20% of precision). An excellent recovery (within 90–110%) between the measured and certified values was also observed for most elements. The recovery rates between 80–120% were obtained for Cr, Mn, As, Nb, Mo, Ag, and U (Table S1). Germanium, Y, Zr, Bi, and most of the rare-earth elements (REE) are not certified by NIST 2702, but we consider the measured values for these elements to be reliable, implying the good agreement of the certified elements.

The contents of Si and Ca in bulk bottom sediments were determined by X-ray fluorescence (XRF) analysis on Spectroskan MAKS-GVM (SPEKTRON, St. Petersburg, Russia) equipped with a vacuum spectrometry chamber (4 crystals LiF200, C002, PET, KAP, in the mode of 40 kV, from 0.50 to 2.0 mA) was used. The accuracy of measurements was controlled while using the SRM SDO-1 (terrigenous oceanic clay) and SDO-3 (carbonate sediment). The accuracy was 6% for Si and 2% for Ca.

Contents of total carbon (TC) and total inorganic carbon (TIC) were determined by the automatic coulometry on a Shimadzu TOC-L-CPN analyser (Shimadzu Corporation, Kioto, Japan). The essence of the method is the oxidation of the carbon compounds contained in the sample at a temperature of 900 °C in the presence of oxygen-containing gas to $CO_2$ followed by subsequent determination of the released $CO_2$ using an infrared detector.

### 2.3.2. The Radionuclide Analysis

Before the analysis, sediment samples were dried in an oven at a temperature of 60 °C to a constant mass followed by homogenization. The content of Cs-137 was determined by a line of 661.6 keV using a gamma-ray spectrometer with a semiconductor detector manufactured from the pure germanium (size of 70 × 25 mm) with Genie 2000 software (Canberra Industries, Inc., Zellik, Belgium). The efficiency of the calibration was examined using standard reference material IAEA-315 and intercalibration standard MAPEP 97 S4 (United States Department of Energy, Washington, DC, USA) [36].

### 2.4. Statistical Data Processing

The methods of multivariate statistical analysis were used to explain the behavior of bulk data of chemical elements in the bottom sediments of the Arctic seas. For that, the data distribution was checked for normality applying the Kolmogorov–Smirnov tests (K–S test), corrected by the Lilliefors test and Shapiro–Wilk (W test). The data that did not pass the normality test was returned to the normal logarithm (logarithmic distribution).

Pearson correlation analysis ($p < 0.01$) was used to determine the relationships between chemical elements content, textural composition (share of the pelitic fraction) of the bottom sediments, and sea depth of the sampling sites. Evaluation of the relationships between the geochemical parameters and identification of the main factors of their variability was performed using principal component analysis (PCA) with alternating "varimax" factors.

This analysis was performed in Statistica 10.0 software package (TIBCO Software Inc., Palo Alto, CA, USA).

### 2.5. Enrichment and Contamination Assessment of Sediments

Assessment of enrichment and contamination levels for studied bottom sediments was conducted by various calculations. The enrichment of sediments in the chemical elements relative to their average content in the lithosphere was estimated by use of $EF_x$ enrichment factor [37]:

$$EF_x = \frac{(C_x/C_{Al})\text{sample}}{(C_x/C_{Al})\text{crust}} \tag{1}$$

where $C_x$ and $C_{Al}$ are the contents of chemical element $x$ and $Al$ in the sample under study and the upper continental crust [38]. The increased content of elements relative to their lithospheric value may result from the natural (hydrogenous and diagenetic) processes, as well as from additional anthropogenic source. The $EF_x$ classification was accepted according to [39].

The degree of heavy metal pollution was estimated using the Geoaccumulation Index $I_{geo}$ calculated as:

$$I_{geo} = log2\left(\frac{C_n}{1.5 \times B_n}\right) \tag{2}$$

where $C_n$ is the content of the $n$ chemical element in the sample, $B_n$—geochemical background value of this element for the Earth's crust (element content in deep-sea clays by [40]). There are seven categories of the $I_{geo}$ index used in this work: $I_{geo} \leq 0$—practically uncontaminated; $0 < I_{geo} \leq 1$—from uncontaminated to moderately contaminated; $1 < I_{geo} \leq 2$—moderately contaminated; $2 < I_{geo} \leq 3$—from moderately to heavily contaminated; $3 < I_{geo} \leq 4$—heavily contaminated; $4 < I_{geo} \leq 5$—from heavily to extremely contaminated; $I_{geo} > 5$—extremely contaminated environment. Applying this index helps to compare the up-to = date heavy metal contents with their pre-industrial level [41].

Assessment of the anthropogenic contamination and ecological risks was calculated for a selection of elements (Cr, Cu, Zn, As, Cd, Pb) and by using single- and multi-element criteria described in Table 2. The selection of these elements is based on their toxic effect on aquatic organisms [42]. The application of different indices allows a more reliable and comprehensive assessment of the heavy metal contamination level in sediments.

**Table 2.** Single- and multi-element criteria of contamination level and potential ecological risk in bottom sediments [42].

| Criteria | Equation | Category | Description |
|---|---|---|---|
| Contamination Factor (CF) | $CF = \frac{C_{0-1}^i}{C_n^i}$, where $C_{0-1}^i$—mean content of elements from at list five sampling sites; $C_n^i$—concentration of elements in the Earth's crust | CF < 1 | Low |
| | | 1 < CF < 3 | Moderate |
| | | 3 < CF < 6 | Considerable |
| | | CF > 6 | Very high contamination |

**Table 2.** *Cont.*

| Criteria | Equation | Category | Description |
|---|---|---|---|
| Potential Ecological Risk Factor (ER) | $ER = T_f^i \times CF$, where $T_f^i$—toxic response factor of element; $CF$—Contamination Factor | ER < 40 | Low |
| | | 40 < ER ≤ 80 | Moderate |
| | | 80 < ER ≤ 160 | Considerable |
| | | 160 < ER ≤ 320 | High |
| | | ER > 320 | Very high risk |
| Degree of Contamination (DC) | $DC = \sum CF$, where $CF$—Contamination Factor | DC < 6 | Low |
| | | 6 < DC ≤ 12 | Moderate |
| | | 12 < DC ≤ 24 | Considerable |
| | | DC > 24 | Very high degree |
| Risk Index (RI) | $RI = \sum ER$, where $ER$—Potential Ecological Risk Factor | RI < 40 | Low |
| | | 80 < RI ≤ 160 | Moderate |
| | | 160 < RI ≤ 320 | Considerable |
| | | RI > 320 | High risk |

## 3. Results

### 3.1. The Grain Size Composition, Total Organic/Inorganic Carbon and Cs-137 Content

The examined sediments of the Kara, Laptev, and East Siberian Seas were represented mainly by clay silts with predominance (75–99%) of pelite fraction (Table 3). In the Kara Sea, the increased proportion of silts (up to 24%) was observed in the Ob Bay (St. 5343) and the Blagopoluchiya Bay (St. 5364), while in the central Kara Sea (St. 5332), the fine sandy-silt sediment prevails. The grain-size composition of the Barents Sea bottom sediments was more diverse: from main pelite (Sts. 6175, 6187, and 6217), and silty-pelite (St. 6196) to silted gravel (St. 6181, 6197). At the same time, sediments of St. 6197, were poorly sorted and consisted of approximately equal parts of pelite, silt, sand, and gravel fractions. In the south-eastern part of the Barents Sea (Sts. 6211, 6215, 5297), there were fine-grained sand sediments, which included well-sorted sand fraction (97%) at St. 6215. It is worth noting, that proportion of granulometric fractions in sediments was estimated without the pebble material; at the same time, the rock fragments coarser than 10 mm were found in the single samples in the Barents Sea (St. 6181, 6197).

**Table 3.** The Cs-137 activity and grain-size composition (%) of bottom sediments. ND—no data.

| Station, Sea | Cs-137 Activity, Bk kg$^{-1}$ | TOC, % | TIC, % | Gravel | Sand | Silt | Pelite |
|---|---|---|---|---|---|---|---|
| 6175, Barents Sea | ND | 2.36 | 0.11 | 0 | <1 | 15 | 85 |
| 6181, Barents Sea | ND | 2.34 | 0.11 | 56 | 7 | 8 | 29 |
| 6187, Barents Sea | ND | 2.35 | 0.07 | <1 | 5 | 19 | 76 |
| 6196, Barents Sea | ND | 1.38 | 0.08 | 1 | 9 | 30 | 60 |
| 6197, Barents Sea | ND | 2.75 | 0.22 | 28 | 24 | 20 | 28 |
| 6211, Barents Sea | ND | 0.25 | 0.06 | <1 | 66 | 24 | 10 |
| 6215, Barents Sea | ND | 0.28 | 0.03 | <1 | 97 | 2 | <1 |
| 6217, Barents Sea | ND | 2.26 | 0.72 | 0 | <1 | 16 | 84 |
| 5297, Kara Gate Strait | 1.2 ± 0.1 | ND | ND | 3 | 55 | 15 | 27 |
| 5302, Kara Sea | ND | 1.18 | 0.12 | ND | ND | ND | ND |
| 5326, Kara Sea | 7.7 ± 0.6 | ND | ND | 0 | <1 | <1 | 99 |
| 5332, Kara Sea | 0.5 ± 0.1 | ND | ND | 0 | 67 | 12 | 21 |
| 5343, Kara Sea | 18.0 ± 0.2 | ND | ND | 0 | <1 | 24 | 75 |
| 5349, Kara Sea | 10.5 ± 0.1 | ND | ND | 0 | <1 | 3 | 96 |
| 5364, Kara Sea | 4.6 ± 0.2 | ND | ND | 6 | 3 | 12 | 78 |
| 5379, Kara Sea | ND | 0.94 | 0.07 | ND | ND | ND | ND |
| 5385, Kara Sea | 2.1 ± 0.2 | ND | ND | <1 | <1 | 5 | 94 |
| 5394, Kara Sea | 5.3 ± 0.6 | ND | ND | 0 | 5 | 5 | 90 |
| 5591, Laptev Sea | ND | 0.200 | 0.18 | ND | ND | ND | ND |
| 5596, Laptev Sea | ND | 0.66 | 0.28 | ND | ND | ND | ND |
| 6475, East Siberian Sea | 1.58 ± 0.5 | ND | ND | 0 | <1 | 17 | 83 |
| 6489, Laptev Sea | 5.93 ± 0.48 | ND | ND | 0 | 1 | 16 | 83 |
| 6505, Laptev Sea | 4.23 ± 0.41 | ND | ND | 0 | 1 | 16 | 83 |
| 6533, Vilkitsky Strait | 6.70 ± 0.68 | ND | ND | 0 | <1 | 14 | 85 |

The concentrations of the total organic carbon (TOC) and total inorganic carbon (TIC) in bottom sediments are listed in Table 3. The TOC content in sediments of the Barents Sea varied 0.75 to 2.75%, while the total inorganic carbon (TIC) had a range of 0.03–0.72%, i.e., 0.25 to 6.0% CaCO$_3$ (from the calculation coefficient 8.33 based on CaCO$_3$/TIC ratio). The TOC content in sediments of the Kara and Laptev Sea is lower, ranging 0.91–1.18 and 0.20–0.66%, respectively.

There are no local sources of technogenic radionuclides in the catchment areas of the Laptev and East Siberian Seas. Thus, the sediments from these water areas demonstrated the lowest Cs-137 concentrations in comparison with the Kara Sea (Table 3). There is also a certain dependence of Cs-137 content on the grain-size composition of the sediments. Thus, minimum Cs-137 activity is observed for samples with high concentrations of gravel and sand fractions (St. 5297 and St. 5332).

*3.2. The Total Element Contents' Distribution*

The total content of 47 chemical elements in the surface bottom sediments of the Arctic seas is listed in Table S2.

To identify the main variability vectors in the elemental composition of the studied sediments, we used the principal component analysis (PCA), which made it possible to simplify this data array (Figure 2). Six significant factors forming the main geochemical

associations of 47 elements in the bottom sediments of the Arctic seas were revealed. The total variance for the six factors was 96.2%. Factor 1, which described 67.4% of the total variance, controlled the distribution of most chemical elements: Be, Al, Ti, Cr, Ga, Rb, Sr, Y, Zr, Nb, Ba, REE, Pb, Th, U, and insignificantly, Zn, Cs, and W; this group includes mainly the lithophile elements. Accordingly, the lithogenic matter was the main mechanism of variability in the chemical composition of the Arctic shelf bottom sediments. Factor 2 (12.9% of total variance) was related to the distribution of Mn, Fe, Co, Ni, Cu, Ge and Mo, as well as, with a degree of conditionality, V and Sb, on which Factor 2 exerted the highest loading but did not reach a significant value of 0.7 (Figure 2). It appears that the hydrogenic processes have a considerable impact on this group of elements, namely a formation of authigenic oxy-hydroxides Fe and Mn, which usually are associated with the fine-grained sediments. A strong negative loading of Factor 2 falls on Si, which was commonly related to the coarse-grained fractions. Factor 3 (5.7% of the total variance) influenced only the distribution of Ca possibly due to the low $CaCO_3$ content (to 6% dry wt.); the rest of the elements were weakly bound to carbonate detritus. The toxic elements (Cd, and As) formed no geochemical association with other elements in the surface sediments of the Arctic seas, being influenced by independent Factors 4 and 6 (4.6 and 2.3% of the total variance, respectively). Factor 5 (3.3% of the total variance) did not exert a significant loading on any element but revealed the strongest loading on Bi.

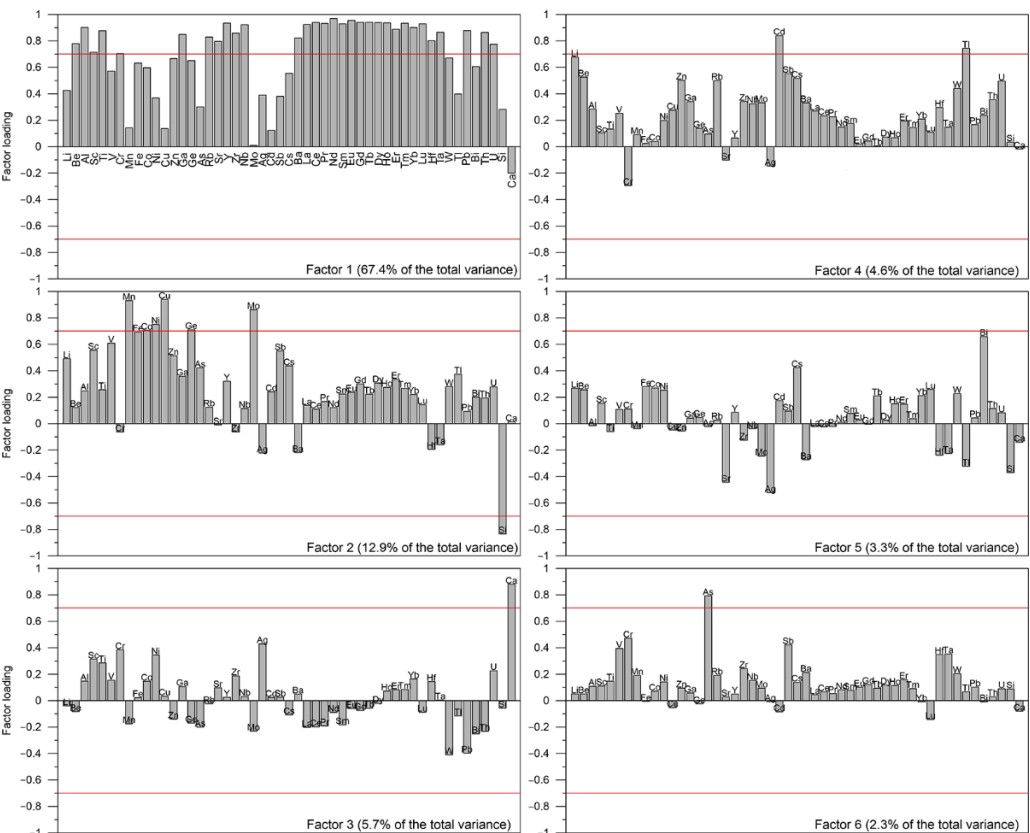

**Figure 2.** Distribution of factor loadings of principle component analysis (PCA) affecting the chemical elements composition of the Arctic Seas bottom sediments for six factors. Statistically significant factor loadings are highlighted in red line (higher than 0.7).

In Figure 3, the spatial distribution of the total contents of 32 elements and the sum of REE in the Arctic seas sediments is displayed. It follows that elevated contents of Mn, Mo, Co, Cu Ge, and V, joint by Factor 2 of PCA, were detected in the Novaya Zemlya Trough (St. 5394), Vilkitsky Strait (St. 6533), and the central Kara Sea (St. 5349) sediments. The maximum content of Zn (213 µg/g) was also found in sediments from the Novaya Zemlya Trough. The sediments of the East Siberian Sea (St. 6475) were characterized by

the maximum content of the most elements (Be, Al, Ga, Rb, Zr, Nb, Cs, Ba, La, light REE, Pb, and Th), which formed the main geochemical association (Factor 1, PCA), as well as Li (67.8 μg/g), Cd (0.56 μg/g) and Bi (0.34 μg/g). The high content of Zn (206 μg/g) was also detected in sediments from the East Siberian Sea. Total Al content was decreased in sandy sediments from some areas of the Barents Sea (St. 6211, 6215), which is attributed to the known Al feature to accumulate in clay minerals constituting a major part of fine-grained instead of coarse-grained sediments. The elevated and maximal Al values, to 8% (the East Siberian Sea), were detected in sediments with a high share of pelite fraction (85–96%). Increased content of lithophilic elements (Factor 1, PCA) was observed in sediments of the Novaya Zemlya Bays (St. 5364 and 5385), Vilkitsky Strait (St. 6533), and the western Barents Sea. Moreover, Ti, Zr, and U were characterized by the maximal content in sediments of the western Barents Sea (Sts. 6175, 6181). Maximum of Cr and Ni content (143 and 72 μg/g, respectively) were detected in the Oga Bay of Novaya Zemlya (St. 5385) in the area influenced by the Ob River runoff (Sts. 5326 and 5349), where the increased Fe content (6.9%) was also found. In the Blagopoluchiya Bay (St. 5364), the higher content of Cd (0.45 μg/g) was determined. The maximal content of As was recorded at St. 5349 in the central Kara Sea. Sediments from the Laptev Sea exhibited the lower content of most elements. The decreased element contents were detected in the coarse-grained sandy sediments in the southeastern Barents Sea (Sts. 6211, and 6215), and the central Kara Sea (St. 5332). A reliable relationship ($r > 0.7$; $p < 0.01$) between the element's content and share of pelite fraction and sampling depth was not detected (Table S3). However, moderate correlation ($r$ 0.3–0.6; $p < 0.01$) between the element contents and the pelite proportion in sediment can be found for Cr, Sr, Zr, Nb, Ag, Ba, some REE, W (inverse) and Mn, Cu (positive).

### 3.3. Level of Contamination and Potential Ecological Risk

The enrichment factor (EF) and the geoaccumulation index ($I_{geo}$) were used to estimate the excess element content in sediments compared to their content in the upper continental crust. The excess content of elements can be attributed to both the anthropogenic pollution and the natural factors (grain-size, organic matter content, diagenetic redistribution). The widespread sediment enrichment was detected only for As (EF 1.7 to 24.2) (Figure 4; Table S4). The Barents Sea sediments, as a whole, were characterized by the decreased degree of enrichment in this toxic metalloid (Figure 4A). The minor, moderate, and moderately severe degrees of As enrichment (EF 1.7–8.1) were common in the sediments of the Barents Sea. At St. 5297 in the Kara Gate Strait, the As EF is 12.1, which corresponded to the severe degree of enrichment. The sediments of the Barents Sea were also characterized by moderate enrichment in Ge. In the western Barents Sea (Sts. 6175–6187), in addition to Ge and As, there was a moderate enrichment in Li, Ti, V, Zr, Mo, Cd, Sb, Pb, and in the case of Li, the enrichment reached moderately severe degree. At Sts. 6197 and 6215, moderate enrichment in Zn (EF 3.0 and 4.7, respectively) was found. In the Kara Sea, at St. 5394 (the Novaya Zemlya Trough), the extremely severe enrichment in Mn (EF 53.3) and Mo (EF to 80.8) was recorded, as well as moderately severe enrichment in Cu (EF 5.2), and Sb (EF 6.2) (Figure 4B). Moderately severe and severe degrees of enrichment in Mn, Mo, and As sediments were also observed in the central Kara Sea (Sts. 5326, 5349), and the Vilkitsky Strait (St. 6533). Similar to the Barents Sea, in the Kara, Laptev, and East Siberian Seas, there was moderate and moderately severe enrichment in Li, Ge, Cd, and Sb. At Sts. 5332, 6476, and 5394, the moderate and moderately severe enrichment of Zn was found (EF 2.9–5.9).

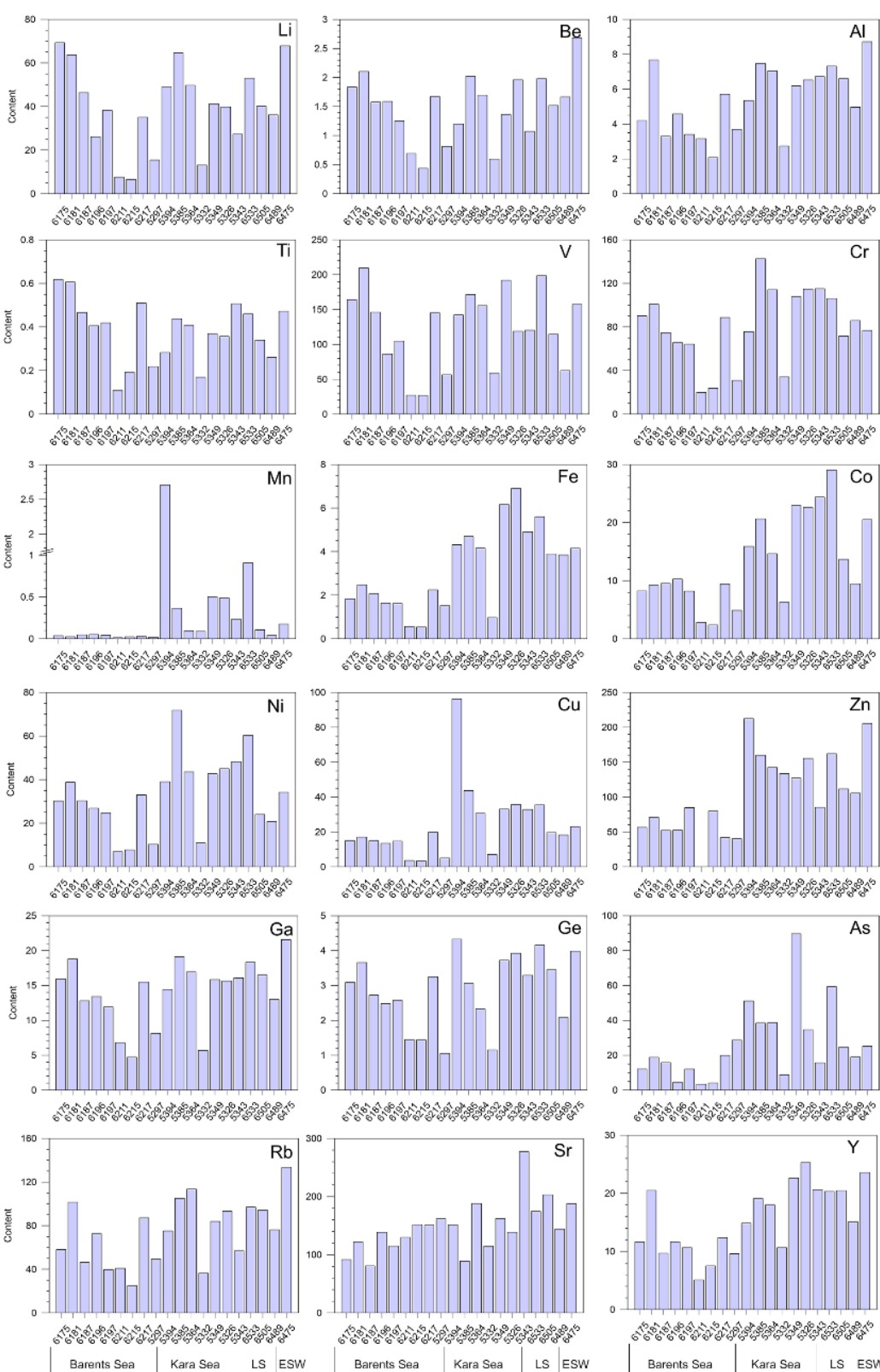

**Figure 3.** *Cont.*

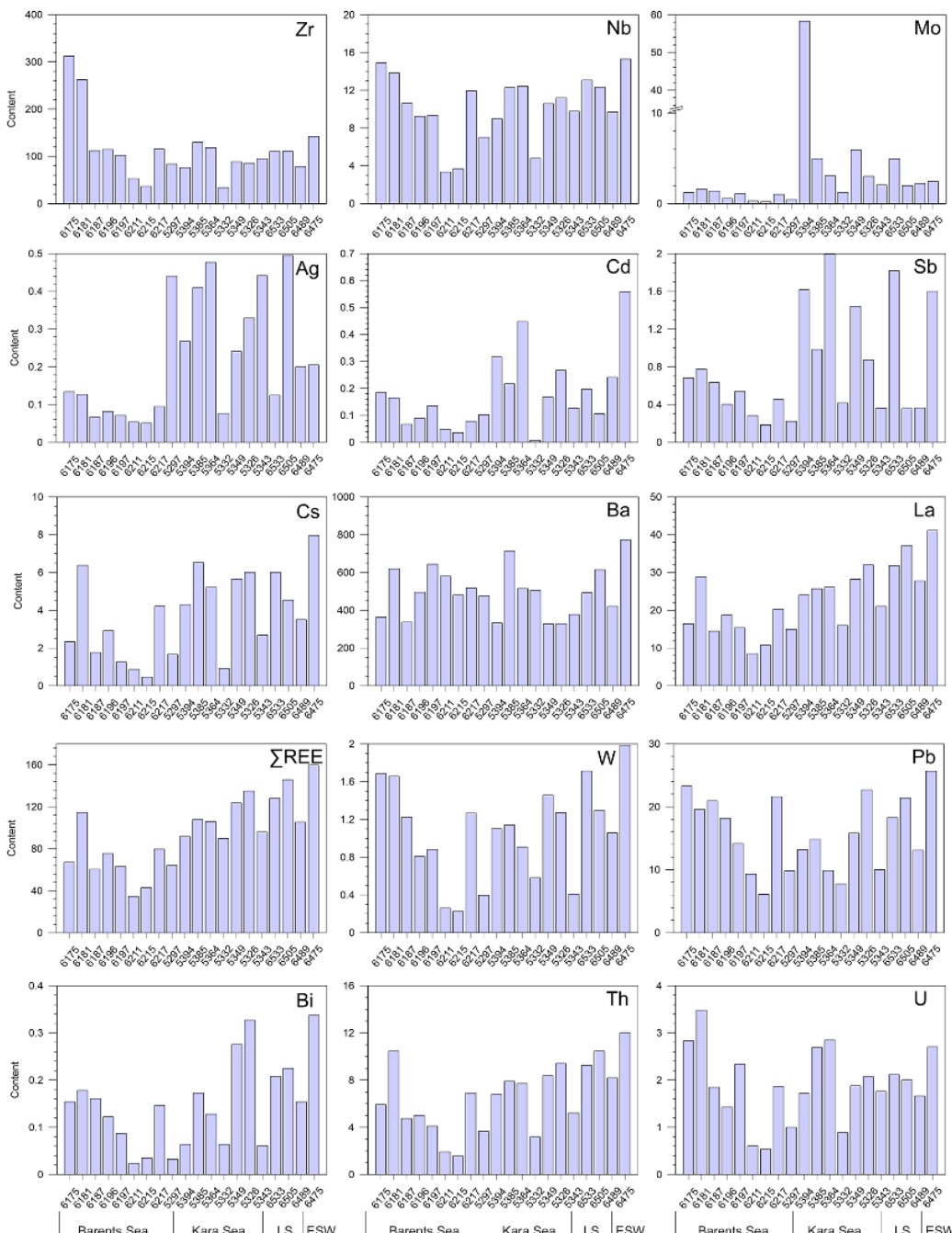

**Figure 3.** Spatial distribution of the total contents (Al, Ti, Mn, and Fe in %, other in μg/g) of elements in surface bottom sediments from the Arctic Seas. LS—Laptev Sea, ESW—East Siberian Sea.

According to the $I_{geo}$ index, for most examined sediments, the excess Mn content (moderately to extremely contaminated) was revealed (Figure 5A,B). In the case of As, only for three stations in the Barents Sea (6181, 6217, and 5297), a moderately contamination (Igeo 1.12–1.62) was detected. Similar to the enrichment factor (EF), the $I_{geo}$ index revealed a moderately contaminatedTi, V, Ge, Zr, and Ag in Barents Sea sediments of the western part (Sts. 6175 and 6181), and at Sts. 6217 and 5297. In the Eastern Arctic (Kara, Laptev, and East Siberian Seas), widespread pollution of Zn, Ge, As, and Ag—from moderately to heavily contaminated, was revealed. The highest $I_{geo}$ index for Cd (1.48) was detected in the East Siberian Sea (St. 6475), where the increased contents of Nb, Hf, and many rare earth elements were found also. A moderately contaminated Cd ($I_{geo}$ 1.16) was also recorded in the Blagopoluchiya Bay of the Novaya Zemlya Archipelago (St. 5364). The extreme

contamination was established for Mo ($I_{geo}$ 5.07) at St. 5394 in the western Kara Sea, where moderate Cu contamination was also detected ($I_{geo}$ 1.68). The moderate contamination of Co, Fe, and V was detected for the Kara Sea at Sts. 6533, 5326, and 5349, respectively.

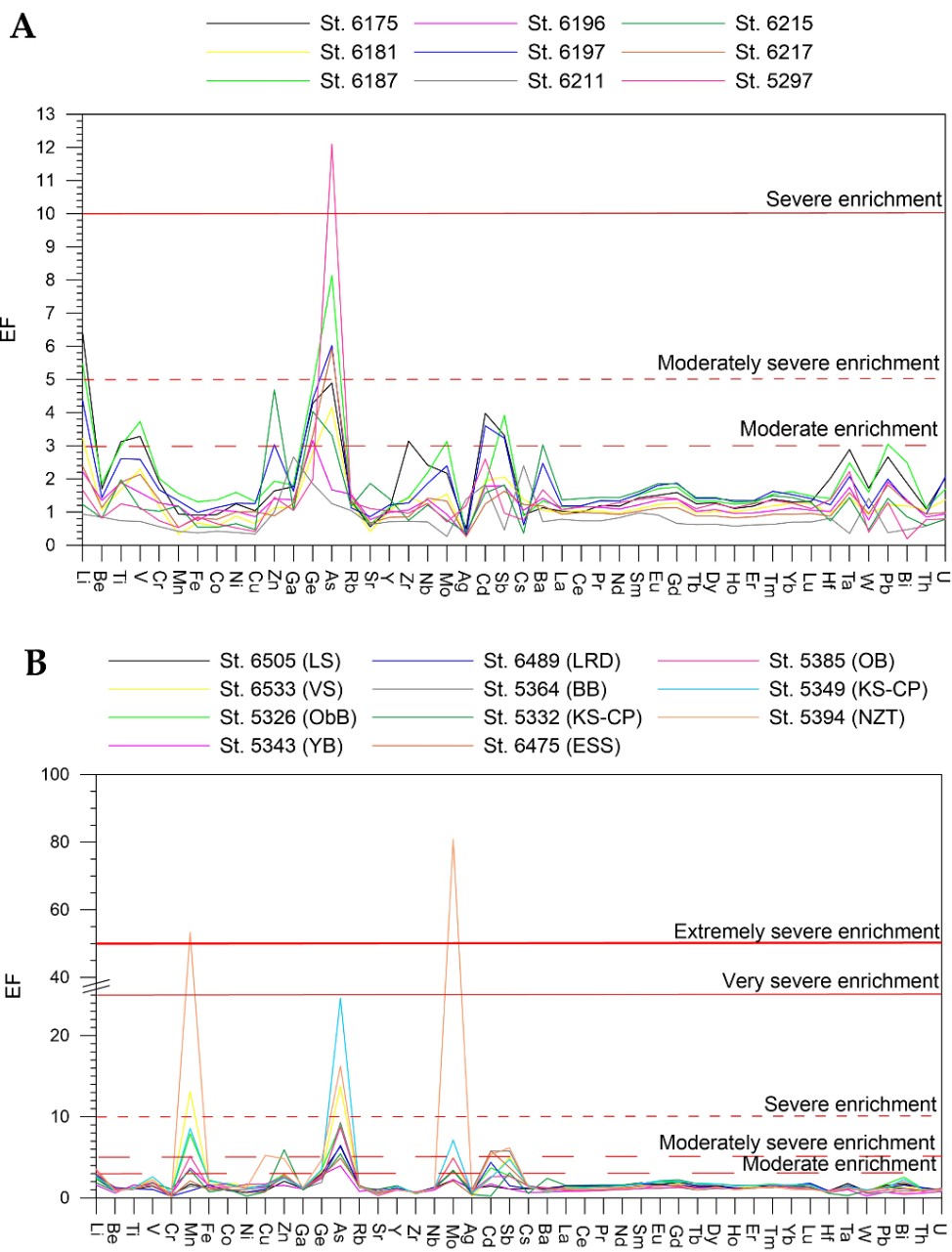

**Figure 4.** Enrichment factor of chemical elements in surface bottom sediments from the Barents Sea (**A**) and Kara, Laptev and East Siberian Seas (**B**). LS—Laptev Sea VL—Vilkitsky Strait; ObB—Ob Bay; YB—Yenisei Bay; LRD—Lena River delta; BB—Blagopoluchiya Bay; KS-CP Kara Sea, central part; ESS—East Siberian Sea; OB—Oga Bay; KS-CP—Kara Sea, central part; NZT—Novaya Zemlya Trough.

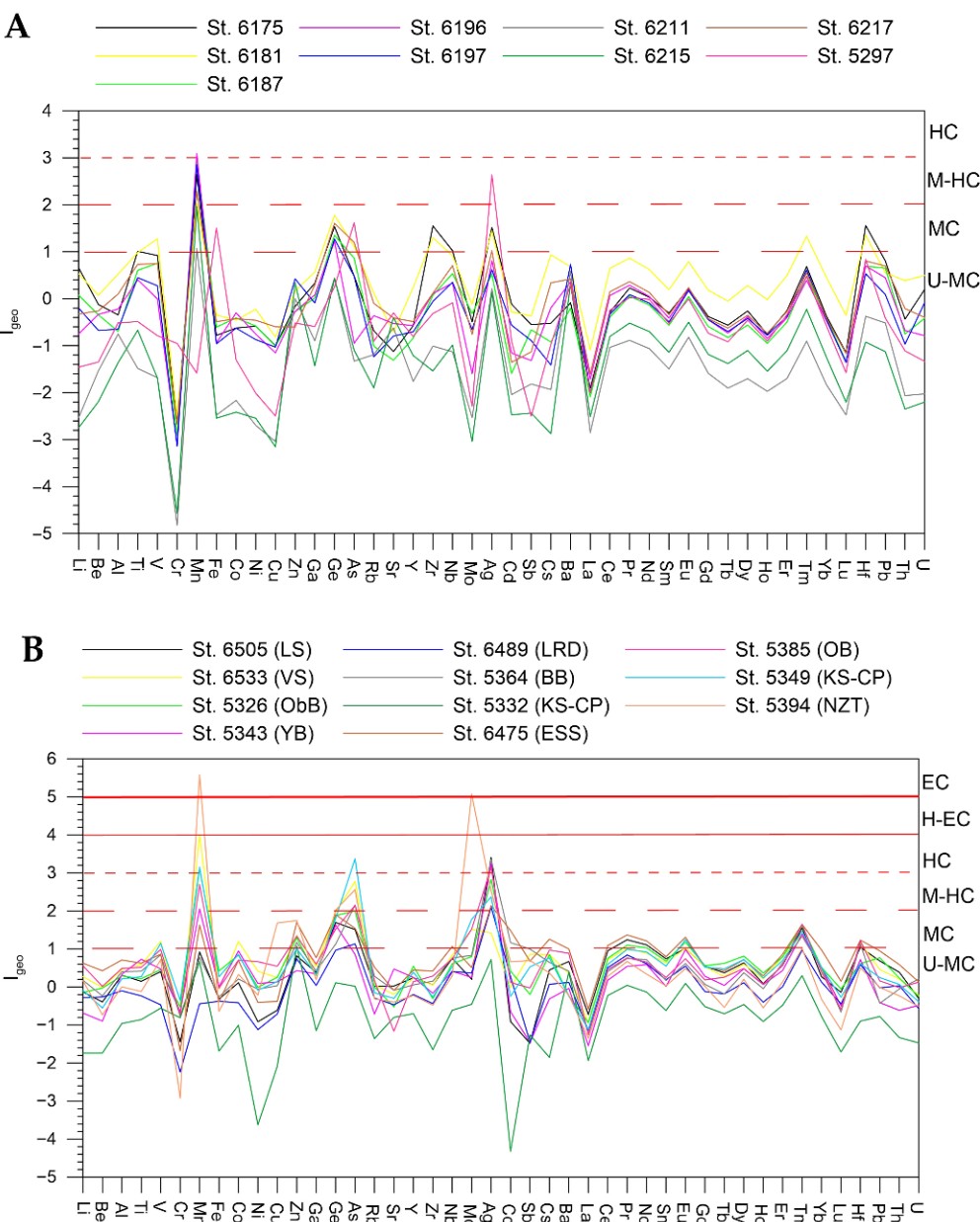

**Figure 5.** Geoaccumulation index of elements in surface bottom sediments from the Barents (**A**), Kara, Laptev, and East Siberian (**B**) Seas. Expansion of sampling site abbreviations is presented in Figure 4. U-MC—Uncontaminated to moderately contaminated; MC—moderately contaminated; M-HC—Moderately to heavily contaminated; HC—Heavily contaminated; H-EC—Heavily to extremely contaminated; EC—Extremely contaminated.

According to the contamination factor (CF), degree of contamination (DC), potential ecological risk factor (ER), risk index (RI) criteria, which estimated the pollution of bottom sediments by six toxic elements (Cr, Cu, Zn, As, Cd, Pb), the least contamination was found for sediments in the Barents Sea (Figure 6). According to the CF criterion, moderate contamination was detected only for As in the north-western Barents Sea (Figure 6A). From the Kara Gate Strait onwards, the moderate As contamination (CF 1.04–2.57) was spreading everywhere, while considerable As contamination was observed in the western and central Kara Sea, as well as in the Vilkitsky Strait (CF 3.41–3.96). Moderate contamination of Zn and Cu was also identified in the western Kara Sea, while moderate Zn contamination was found in the East Siberian Sea. In turn, the contents of Cr, Cd, and Pb were everywhere characterized by low contamination (CF < 1). According to the DC criterion, sediments of

the western Kara Sea (Novaya Zemlya Trough) and Vilkitsky Strait revealed a moderate degree of contamination in As due to its high content in these parts areas (Figure 6B). The other areas were characterized by a low degree of contamination (DC < 6). According to a potential ecological risk factor (ER), all elements exhibited low risks (ER < 40) in examined sediments (Figure 6C). In accordance with the risk index (RI), all studied areas of the Arctic seas demonstrated a low degree of ecological risks (Figure 6D).

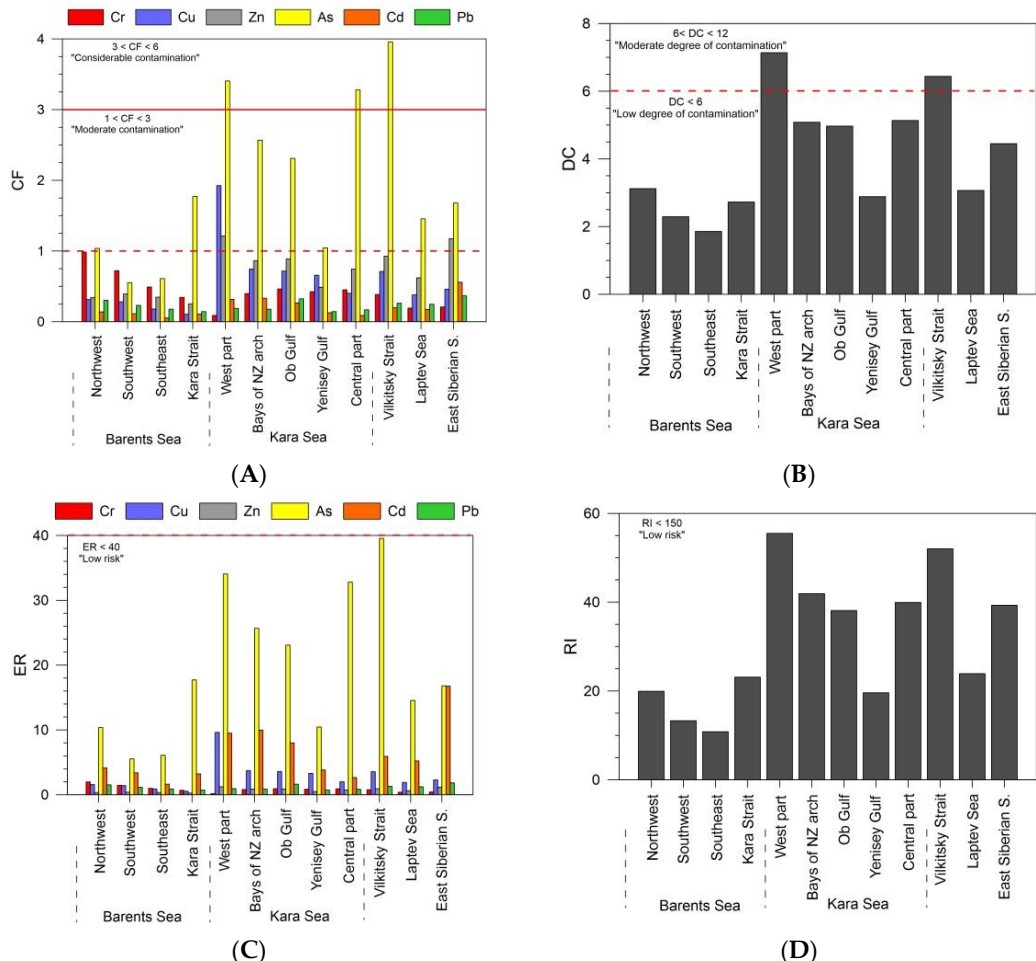

**Figure 6.** The criteria of contamination level and potential ecological risk of toxic elements in surface bottom sediments from the Arctic Seas: (**A**) Contamination factor CF; (**B**) Degree of contamination DC; (**C**) Potential ecological risk factor ER; (**D**) Risk index RI. While calculating these criteria, a mean content of 6 toxic elements was used from several (up to 5) nearby sites. In our case, 20 sampling sites were combined into 12 locations.

## 4. Discussion

### 4.1. Controlling Factors of Elements Distribution

In the Barents Sea sediments, a relationship between the distribution of mineral complexes and the main sources was noticed, while the geochemical characteristics of sediments were naturally related to the variety of mineral complexes marked with structural peculiarities of the minerals included [22]. A characteristic feature of sediments from the eastern Barents Sea (the Novaya Zemlya Archipelago) is a predominance of clay minerals (a combination of high chlorite content with muscovite, sericite, illite), and the elevated contents of many trace elements were detected in the pelite fraction with an abundance of the clay minerals. On the contrary, in the shallow and nearshore sediments, quartz and feldspars prevailed, and decreased contents of most trace elements were detected [22].

The examined sediments contain low quantities of organic and inorganic carbon (Table 3), which is typical for the Arctic regions [23,43]. The pelite material (grain size < 0.01 mm) predominates in most sediment samples, which leads to an increase in their sorption capacity and consequently to the higher concentrations of trace elements (Table S2). However, a weak effect of grain-size composition on the element contents (Table S3) was revealed. The main factor determining the geochemical characteristics of Arctic sediments seems to be the composition and textural features of minerals [22]. A slight dependence on the pelite proportion was observed only for Mn and Cu (Table S3), which was due to their predominant fine-grained carriers (oxyhydroxides, organic matter) [44]. On the contrary, the predominance of the coarser silt-sandy material in four samples (Sts. 6211, 6215, 5297, and 5332), and even more so gravelly material (Sts. 6181 and 6197), contributes to a decrease in the element contents.

The variability of the main association of elements (Be, Al, Ti, Cr, Ga, Rb, Sr, Y, Zr, Nb, Ba, REE, Pb, Th, U, and insignificantly, Zn, Cs, and W) detected by PCA (Factor 1; 67.4% of the total variance) is related to the distribution of the terrigenous fraction. In terms of the grain-size composition, many of these elements (Cr, Sr, Zr, Nb, Ag, Ba, REE, W) are prone to enrichment in the coarser-grained material (Table S3). Consequently, the abundance of these elements in the sediments studied can be explained by the prevalence of heavy minerals [45].

The second geochemical association of Mn, Fe, Co, Ni, Cu, Ge, Mo, and, to a lesser extent, V and Sb (PCA, Factor 2), is identified by the redox-sensitive Mn and Fe, which are involved in the diagenetic process [46]. At the interface with the oxygenated seawater, enrichment in insoluble forms of $Mn(IV)$ and $Fe(III)$ occurs due to oxidation of the reduced form of $Mn(II)$ and $Fe(II)$, supplied with the diffuse flux from the underlying reduced sediment layers, as well as due to the advective inflow of $Mn(II)$ and $Fe(III)$ with bottom currents [19,47,48]. As a result, under conditions of low biological productivity and low rates of sedimentation in the Arctic seas, brown surface layers of bottom sediment were formed, enriched with oxidized forms of Mn and Fe, which bind some heavy metals (Cu, Co, Ni, Mo) [48,49].

Despite the Cu distribution in the marine sedimentation being commonly controlled by its association with organic matter [50–52], Cu is found in this study among the elements bound to Fe and Mn oxyhydroxides. Similar results were found in the study of the Svalbard Fjord sediments, where Cu showed a very strong correlation with Mn (r = 0.99) [53]. It is likely that in the Arctic seas, characterized by a low content of organic matter, mobile Cu was scavenged on the Mn oxyhydroxides.

The remaining 4 significant PCA factors are related to the individual distribution of Ca, Cd, Bi, and As. According to the correlation matrix (Table S3), Cd exhibits only a moderate correlation (r = 0.3–0.7; *p* > 0.01) with some elements Li, Be, Al, Zn, Ga, Rb, Zr, Nb, Cs, Er, Yb, W, Bi, Th and U. Thus, Cd and some lithophile elements can be incorporated into the sea ice and aerosols as part of fine-grained clay material followed by their discharge and deposition [54,55]. There are also the closest similarities in the spatial distribution of Cd and Zn (r = 0.5; *p* > 0.01). Therefore, such a behavior of Cd, as well as Zn, can be explained by their uptake by marine plankton [56–58], or the presence of Cd in the form of impurities in the Zn minerals [59]. One of the explanations for Cd accumulation in sediments of the East Siberian Seas could be their adsorption bind to Fe and Mn oxyhydroxides [23].

A characteristic feature of As is its occurrence in seawater in the negatively charged form ($AsO^{2-}$, $AsO_4^{3-}$, $HAsO_4^{2-}$, $H_2AsO_3^{-}$) unlike the other examined elements represented by positively charged ions. The As oxy-anions can bind to positively charged particles of the sediments by weak electrostatic forces [60]. That implies that the sorption mechanisms for As are different from those for other elements, and that can explain the As separate geochemical association (PCA, Factor 6). In turn, the geochemical behavior of Bi can be explained by its high adsorption capacity in the bottom sediment, and it is likely bound to colloidal organic substance [61].

In general, variability in the elemental composition of the bottom sediments in the Arctic seas is approximately 70% controlled by the content of the lithogenic material (PCA). This corresponds to the general regularities of the Arctic basin sedimentation which is controlled by the terrigenous material supply by the river runoff, coastal abrasion, eolian routes, and transport by sea ice and icebergs [62]. Among the hydrogenic processes, the formation of Fe and Mn oxyhydroxides (about 13% from PCA data) has the greatest impact on the element variability in sediments.

*4.2. Contamination and Ecological Risks*

Before discussing the identified levels of enrichment/contamination of the Arctic seas' sediments with chemical elements and associated environmental risks, we will compare our data on the total concentrations with the earlier studies. In Table S5, the content of Li, Al, V, Cr, Mn, Fe, Co, Ni, Cu, Zn, As, Mo, Cd, and Pb in surface sediments of the Barents, Kara, Laptev, and East Siberian Seas, obtained over the past 25 years, are listed. In the Barents Sea, we detected a lower level of heavy metals (Cr, Co, Cu, Ni, Zn, Pb) compared to the data [21]. In our study, minimum levels of elements in the sediments of the Barents Sea were associated with an increased proportion of the gravel-pebble and sandy grain-size fractions. Transport and unloading of coarse-grained material by seasonal ice and icebergs is a characteristic feature of sedimentation in the Barents Sea [29]. We also made a comparison of the average contents of examined elements with those deposited in the pre-industrial periods, based on our study from the sediment core in the Barents Sea [63]. We found that the current content of Al, V, Cr, Mn, Fe, Co, Ni, Cu, Zn, As, Mo, Cd, and Pb in the surface sediments of the Barents Sea, in general, corresponds to their background (pre-industrial) level.

For element contents in the Kara, Laptev, and East Siberian Seas sediments, there was a good correspondence of our data and the literature data. In a comparison with our study for the Kara Sea shelf area, very high levels of Fe (up to 19%), Co, Ni, and As (up to 99, 119, and 710 µg/g, respectively) were found in the sediments of the St. Anna Trough [64]. Aerosol transport from enterprises of the Kola Peninsula, the Urals, and Siberia (thermal power plants, mining, metallurgy, pulp, and paper industry and others), as well as the Ob River runoff, are assumed to be the different sources of heavy metals and As into the Arctic seas. It is noted that in the Arctic seas, pollutants can be transported over long distances by currents, being fixed on the sea ice particles [55]. Carriers of metals and As in the sediments in the St. Anna Trough are the organic matter and Fe oxyhydroxides, carried by bottom currents along the continental slope to the deeper sea area [64]. Comparability of our study with studies conducted in the Kara and Laptev Seas in the 1990s suggests that the anthropogenic load on the Arctic environment in terms of pollution with heavy metals has not increased over the past 25 years.

Despite a low content of chemical elements in the examined sediments of the Barents Sea, a moderate and moderately severe (in the case of Li) enrichment in some elements (Li, Ti, V, Ge, As, Zr, Mo, Cd, Sb, and Pb) was found in the northwestern part of this sea. The presence among these elements of lithophylic ones (Ti, Ge, and Zr) suggests a natural origin of the elements' increased levels in the northwestern Barents Sea. A possible supply of pollutants via advection of water masses in this part of the Barents Sea is doubtful, given that it is influenced by the polar Arctic waters transported from the central Arctic to the southwestern direction [24]. The technogenic origin of the moderate enrichment of Zn (EF 3.0–4.7) of some areas of the Barents Sea (St. 6197 and 6215) is not confirmed. In the surface sediments of the Barents Sea, the background content of Zn is ubiquitous, except for the narrow coastal area of the Kola Peninsula [21]. At the same time, our study on Zn content (42–85 µg/g) is much lower than the range of 150–241 µg/g [21]. According to other criteria (CF, DC, ER, RI), no pollution with Zn of the Barents Sea sediments was detected.

A moderate and moderately severe enrichment in Cd (EF 3.6–5.8) of some samples (St. 6175, 6197, 5326, 5364, 5394, 6475, 6489) is combined with a high Zn content in these sediments (moderate and minor degree of EF). It presumably indicates the presence of Cd

in the form of impurities in the Zn minerals [59], and as a consequence, suggests its natural origin. Moderately contamination with Cd according to the $I_{geo}$ index is observed only in sediments of the East Siberian Sea and the Blagopoluchiya Bay of the Novaya Zemlya Archipelago, where Cd content reaches a maximum (0.56 µg/g) among the areas studied. The high Cd content (up to 0.59 µg/g) in sediments of the eastern part of the East Siberian Sea was found earlier, which was explained by the massive development of phytoplankton in this part of the sea, due to the entering productive waters mass from the Bering Sea [23]. It should be noted that other contamination criteria (CF, ER) did not identify Cd pollution in recent Arctic sediments (Figure 6).

As discussed above, the Mn enrichment/contamination of the sediments of the Arctic seas is related to the deposition of Mn oxyhydroxides at the sediment–water interface, scavenging some elements (Mo, Cu, and Ge) [48]. The extremely severe degree of enrichment of Mn and Mo (EF > 50) of the western part of the Kara Sea (the Novaya Zemlya Trough) stands out, which must be provided with a necessary amount of the dissolved Mn (II) being oxidized to Mn (IV). In this case, it is worth considering the external sources of Mn inflow into the Arctic seas, among which the river runoff is distinguished. For small rivers draining the West Siberian lowland, a significant enrichment in Mn has been revealed (on average 5-fold) compared to the average value of the rivers of the World [65]. The Mn enrichment of river suspension in the Ob and Yenisei is also emphasized [9,66]. The catchment area of the West Siberian Rivers includes many lakes and bogs that are the source of Mn(II), which is subsequently oxidized and enriches the river suspension with an oxy–hydroxide form [65]. It is in the sediments of the Kara Sea and the Vilkitsky Strait (Sts. 5326, 5349, 5533, 5394), where the influence of the river runoff of the Ob and Yenisei affects, that we observed the highest total Mn contents (0.49–2.7%) among the studied marine areas. The Novaya Zemlya Trough (St. 5394), which, although not under the direct influence of the runoff of the Ob and Yenisei Rivers, but periodic penetrations of freshwater lenses into the western part of the Kara Sea take place [67]. Moreover, the agitating fluffy layer with the Mn-enriched suspension is carried by the bottom currents over considerable distances and accumulates in the seafloor depressions along with the other finely dispersed material [19].

The high content of As (51.1–89.6 µg/g) was found in the Kara Sea and the Vilkitsky Strait sediments (Sts., 5349, 5394, 6533), enriched in Mn (0.5–2.7%). However, following the PCA, As exhibits no significant relationship with the group of elements associated with the Mn and Fe oxyhydroxides. According to the correlation matrix, the distribution of As is also significantly correlated only with the distribution of V and Sb (r = 0.78; *p* > 0.01), although a moderate correlation (r = 0.3–0.7; *p* > 0.01) is observed with most of the other elements (Table S3). It seems to be a proportional differentiation of the As occurrence forms. Arsenic is partly present as impurities in lithogenic material and partly as an adsorbed anion complex on the surface of positively charged Fe oxyhydroxides [68]. The formation of its geochemical association (PCA, Factor 6) may be related to anthropogenic sources of As, which could be the combustion products of peat and coal, as well as the weathering of coal-bearing strata of sedimentary rocks characterized by a high content of arsenic [69,70]. About 30 years ago, As contamination of the Pechora Sea sediments (up to 308 µg/g) was found and was attributed to the As emissions during nuclear weapons tests at the military test sites of Novaya Zemlya Archipelago [7]. This conclusion was confirmed by a strong correlation (r = 0.93) between the total As contents and the technogenic radioactive isotope Cs-137, which serves as a product of nuclear fission during the test explosions. Currently, there was no correlation between As and Cs-137 in sediments of the Barents, Kara, Laptev, and East Siberian Seas ($r^2$ = 0.014). Thus, if current arsenic contamination of Arctic sediments is to be expected, it is not related to radioactive contamination. However, sediments of the western and central Kara Sea and the Vilkitsky Strait revealed a considerable degree of As contamination, corresponding to the CF criterion, and a moderate degree of contamination, according to the DC criterion. At the same time, according to the single-element criterion (ER) and the multi-element criterion (RI) of ecological risks, the sediments of all examined

areas were characterized by a low degree of risk. To date, a low degree of ecological risks was also identified by the RI criterion in the Laptev and East Siberian Seas' sediments [23].

In general, the Arctic sediments studied were characterized by a low degree of heavy metal contamination except for As. To better identify the sources of the As contaminated areas of the Arctic seas, the speciation of this toxic metalloid is necessary to examine in more detail in future studies.

### 4.3. Cs-137 in the Arctic Seas' Sediments

Radionuclides are actively involved in biogeochemical cycles, accompanied by their redistribution in aquatic ecosystems wherein their removal and secondary accumulation on geochemical barriers occur [71,72]. The specific activity values for Cs-137 in the surface layer of the examined sediments vary within the limits of 0.5–18 Bq/kg (Table 3). In general, these levels are comparable to the average background values of technogenic activity characteristic of the Arctic seas [32,36,73].

For the geochemical behavior of cesium in a marine environment, active participation in ionexchange and sorption interactions with clay particles is characteristic. In addition, cesium can selectively penetrate the interlayer spaces of layered clay minerals (illite, smectite, montmorillonite). In our study, the lowest levels of Cs-137 activity (1.2 and 0.5 Bq/kg) were observed for two samples (Sts. 5297 and 5332, respectively) with high concentrations of coarse grain-size particles (sum of gravel and sand), constituting a total of more than 70% of the total sediment. The coarse particles (>0.1 mm) are commonly not able to retain this radionuclide, resulting in a decrease in the specific content of Cs-137 to a value which is below the global average level characteristic of this region. The minimum content of Cs-137 (1.58 Bq/kg) was found in the sediment from the central part of the East Siberian Sea (St. 6475), the farthest from the main potential migration routes of technogenic radionuclides. This sea might be considered the cleanest one compared to other studied ones, constituting a conditional raw of the technogenic contamination's degree in the direction from the Kara and Laptev Seas towards the East Siberian Sea. This pattern agrees well with [74].

The highest levels of the Cs-137 specific activity (Bq/kg) were detected at St. 5326—7.7, St. 5343—18, St. 5349—10.5 in the sediments of the Ob and Yenisei Rivers estuaries. The estuarine areas of the great Siberian Rivers are zones where the powerful marginal filters in which the processes of coagulation of colloidal substances and suspended matter, including human-made contaminants, are actively operated [75]. The river-run supply of a huge amount of suspended particulate matter and its deposition in the river-sea mixing zone plays an important role in the accumulation of Cs-137 in surface bottom sediments.

The content of radionuclides in sediments is largely affected by the forms of their supply directly from the source. In this regard, a significant difference in the degree of fixation by sediments of radioactive isotopes and their stable analogs is often observed [76]. In particular, this is manifested in a different mechanism of accumulation, in different residence times of stay in the water column, etc. The literature data agree well with our obtained absence of correlation between the specific activity of Cs-137 and the content of stable cesium in sediments ($R^2 = 0.014$).

In general, a correlation between Cs-137 and grain-size composition can be noticed. In those samples where the lowest contents of the pelite fraction were found, the content of Cs-137 was minimal. The rest of the sediment samples contained more than 75% wt. pelite. It is probable that variations in content of the Cs-137 were attributed to a presence of potential local sources of pollution, rather than the content of a fine-grained fraction. In the literature, a general tendency for cesium affinity for clay minerals was described, but there is no clear direct correlation with grain size [77]. The relationship between content of Cs-137 and total organic carbon (TOC) was not found for our samples, due to the lack of statistical data on TOC. In addition, the examined Arctic sediments have very low TOC content. A similar relationship could be traced to soils with typical high TOC content.

## 5. Conclusions

The total element content, including radioactive isotope Cs-137, in the sediments of the shelf zone of the Barents, Kara, Laptev, and East Siberian Seas was investigated. According to the PCA, approximately 70% of the elemental variability in the bottom sediments of the Arctic seas is associated with the lithogenic material, which controls the distribution of the most the examined elements (Be, Al, Ti, Cr, Ga, Rb, Sr, Y, Zr, Nb, Ba, La, Ce, Pr, Nd, Sm, Eu, Gd, Tb, Dy, Ho, Er, Tm, Yb, Lu, Pb, Th, U, and unreliably Zn, Cs, and W). Among the hydrogenic processes, the influence of the authigenic Fe and Mn oxyhydroxides factor is distinguished, which can explain about 13% of the total variance of data. This factor affects the variability of Mn, Fe, Co, Ni, Cu, Ge, and Mo, and insignificantly V and Sb. In this work, the toxic elements Cd and As form no geochemical association with other elements, and they are influenced by independent components. Accumulation of Cd is likely associated with planktonic organic matter or the presence of Cd in the form of impurities in the Zn minerals. While a distinctive feature of the metalloid As, which occurs in the oxy-anions form in seawater, determines the different mechanism of its adsorption on the sediment-forming components. In spite of this, an impact of anthropogenic activity on the As uptake to the Arctic environment should not be excluded.

A comparison with the studies conducted in the Barents, Kara, and Laptev seas over the 1990s indicates no increase in the anthropogenic heavy metal load in the recent decades. In some areas, the enrichment/contamination, detected by the EF and $I_{geo}$ indices, for a number of elements (Li, Ti, V, Mn, Cu, Ge, As, Zr, Mo, Ag, Cd, and Sb) is likely caused by natural processes, provided by diagenetic Mn and Fe redistribution in the case of the mobile elements, and features of the mineral composition of sediments in the case of the inert elements. The exclusion is As, for which considerable and moderate degrees of contamination of the western (the Novaya Zemlya Through) and the central Kara Sea, as well as the Vilkitsky Strait, was detected (by contamination factor CF, and degree of contamination DC). However, according to the Potential Ecological Risk Factor ER and Risk Index RI, all the examined areas exhibited a low degree of risk. The As increased association with radioactive contamination caused by testing the nuclear weapons on Novaya Zemlya in the 20th century has not been confirmed to date due to the lack of correlation between As and human-made Cs-137. However, the presence of an anthropogenic source of As associated with the combustion of peat and/or coal is theoretically possible. Therefore, an investigation of the influence of the anthropogenic sources on the As content in the Arctic sediments, accompanied by an analysis of the occurrence forms of this toxic metalloid, should be given more attention in future studies.

Data on the specific activity of Cs-137 in the surface layer of modern sediments of the Russian Arctic Seas correspond to the background average values characteristic for these regions. The least polluted with technogenic radionuclides area, taking into account the grain-size composition of sediments, is the central part of the East Siberian Sea (1.58 Bq/kg). Maximum levels of radioactivity (up to 18 Bq/kg) were recorded in the Ob and Yenisei Rivers' estuaries. The content of Cs-137 is extremely low (0.5–1.2 Bq/kg) in the coarse-grained sediments, which consist mainly of sand and gravel. An absence of a positive correlation of the contents of the stable Cs with the specific activity of technogenic Cs-137 allows us to conclude that their different geochemical behavior resulted from their fundamentally different origin.

**Supplementary Materials:** The following supporting information can be downloaded at: https://www.mdpi.com/article/10.3390/min12030328/s1, Table S1: Some characteristics of the analytical determination of the elements examined; Table S2: Total element contents in bottom sediments of the examined Arctic sediments; Table S3: The correlation matrix for the content of the chemical elements, the share of the pelitic fraction in the bottom sediments and sea depth of the sampling sites from the Arctic seas; Table S4: Distribution of EF, $I_{geo}$ and criteria of contamination level and potential ecological risk in the bottom sediments of the Arctic seas; Table S5: Range of element concentrations in surface sediments from the Barents, Kara, Laptev, and East Siberian Seas according to the data from various investigations.

**Author Contributions:** Project idea, L.L.D. and D.F.B.; sampling and onboard preparation, D.F.B. and A.V.T.; grain-size analysis and data interpretation, T.N.A.; radinuclides' analysis and interpretation, A.V.T.; chemical analysis and data processing, D.F.B. and D.P.S.; writing, D.F.B., L.L.D., A.V.T.; funding acquisition, L.L.D. All authors have read and agreed to the published version of the manuscript.

**Funding:** This research was carried out in the framework of a State Assignment of Ministry of Science and High Education, Russia, the program no. FMWE-2021-0006 (IO RAS) and the program no. 0137-2019-0010 (GEOKHI RAS); obtaining of field material and its treatment was supported by the Russian Science Foundation (https://www.rscf.ru), project no. 19-17-00234.

**Data Availability Statement:** Not applicable.

**Acknowledgments:** The authors are grateful to the crews and all participants of expeditions on the Research Vessel "Akademik Mstislav Keldysh" for the invaluable support before and during expeditions.

**Conflicts of Interest:** The authors declare that they have no known competing financial interests or personal relationships that could have appeared to influence the work reported in this paper.

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
