# Peer review of "The Features of Distribution of Chemical Elements, including Heavy Metals and Cs-137, in Surface Sediments of the Barents, Kara, Laptev and East Siberian Seas"

_minerals, doi:10.3390/min12030328_

Round 1

Reviewer 1 Report

The paper examines geochemical fractions (including heavy metals and radionuclides) in surface sediments of four areas of the Artic region. In spite of the promising objectives, the study suffers from a series of fatal failures. As a result, their data interpretation and further conclusions might be in accurate or even erroneous. I suggest the authors discard the results of chemical speciation. In addition, the manuscript should be shortened and the number of figures and tables reduced as much as possible. These and other considerations are detailed below:

  • Lines 2-4. The title should be changed in case data from geochemical fractions are eventually discarded as suggested.
  • Lines 15-18. The objective number 3 is missing here.
  • Lines 67-70. Check grammar and coherence. Relate this point to the previous paragraph, including some references to back up the arguments.
  • Lines 71-103. This part need to be re-structured completely. It is worth mention there are many sequential extraction procedures in the literature (check the list of references at the end of this review). All of these can provide information about the bioavailability and ecotoxicity of elements. The selection of only two geochemical fractions is, therefore, purely arbitrary. Start pointing out the advantages of using extraction procedures to evaluate the pollution in aquatic environments. Secondly, highlight the main findings of this type of studies in the Arctic region. And finally, the authors need to give reasons and motivations behind their study.
  • Line 79: "... geochemical speciation /partitioning/ fractionation/etc."
  • Lines 182-183. Oven-drying is not suitable for chemical speciation because it alters natural conditions of sediment (Baeyens et al., 2003; Huang et al., 2015). In fact, it is totally unacceptable when analysing anoxic samples like these (Hirner, 1992; Larner et al., 2008). This may lead to erroneous conclusions (see my remarks below).
  • Lines 206-215. Admittedly, a quality control of the results can be tested by analysing standard reference materials. However, it should be stressed that recovery rates must be close to 100% whereas precision error and accuracy (bias) should be as low as possible. The recovery of a sequential extraction is also determined by comparing the sum of different fractions with a separate determination of total elements. Other studies include repeatability or reproducibility of the measurements. Since the terms recovery, accuracy or precision are not the same, the authors must explain how were obtained. I think the works of Baeyens et al. (2003), Jafarabadi et al. (2017), Snape et al. 2004, Sutherland (2000) and Sutherland and Tack (2003) could take as an examples for making calculations. Check and clarify.
  • Lines 228-238. Put this part after the description of the sequential extraction procedure. Include references.
  • Lines 240-242. The extraction with cold 1 M HCl is more suitable for the quantification of mobile and bio-available fraction (Choi et al., 2012; Snape et al., 2004) due to the potential dissolution of clays and other aluminosilicates (see my observations below). Include the temperature of the ultrasonic bath.
  • Lines 244-247. These results are missing. Total contents seem to correspond to the results obtained by the sequential extraction procedure only.
  • Lines 248-267. The method presented here is quite different to the reference number 43. Explain the modifications and include references to support any such changes.
  • Lines 274-278. Some of the reported concentrations are found below detection limit (see Table S1). These concentrations should be considered zero for the statistical analyses. Please, clarify if this was taken into account.
  • Lines 301-303. Re-phrase as follows: “... was calculated for a selection of elements (Cr, Cu, Zn, As, Cd, Pb) and by using the multi-element criteria ..." What is the underlying reason of choosing only these six elements? Was it based on a previous inspection of EF and Igeo values? Explain this matter on the text.
  • Line 324. Move the last sentence to Discussion section and include some references to support such statement.
  • Lines 327-28. The concentrations of most elements are in the range of ppb. However, the authors did not mention if clean-up instrumental procedures were carried out during handling and analysis of samples (for instance, utilization of acid-washed flasks). This is vital for the analysis of trace elements in soil and sediment samples (Mclaughlin et al., 2000) Point out this matter in M&M.
  • Lines 328-329. Include the value obtained by the single total digestion as well.
  • Lines 331-332. Again, clarify if the total contents correspond to the sum of the two fractions or were obtained in a separate digestion. It is more adequate to display the latter in Fig. 2.
  • Lines 357-358. It is unclear whether the total contents correspond to the sum of the two fractions (same as in Table S1) or are taken from a separate single digestion. Clarify. On the other hand, the observed differences in the elemental contents of the two fractions are difficult to see by this manner. Since this information is already given in Tables 6 and S1, it would have been advisable to show the % of the two fractions (taken from Table S2) instead of their concentrations in Fig. 3. However, these data must be removed from the study due to fatal failures during the preparation and analysis of samples (see my comments on it). If so, the PCA should be re-calculated as well.
  • Line 405. This section must go before. It turns out that the separation into the four groups has been already used in Fig. 3 (section 3.1.1). However, nothing was said until here.
  • Lines 408-410. This statement is a platitude unnecessary to mention. Remove. The authors should indicate the more remarkable features of the four regions studied (Barents Sea, Kara Sea and Laptev and East Siberian Seas) more than discussing the differences between individual stations. I also note the lack of a description of potential differences related to sea bottom depth of the sampling sites.
  • 442-444. The presence of substantial quantities of reactive Al (7-31%; Table S2) reveals solubilisation of aluminosilicates during the extraction with 6M HCl. Consequently, this extractant is not valid for the evaluation of bio-available fraction of sediment. The authors should have applied more diluted concentrations of this acid (e.g. 1M HCl; Snape 2004, Choi 2012). Nevertheless, the method of samples preparation might be the underlying cause. It is known that the air- and oven-drying of sediment samples is unsuitable for heavy metal partitioning because both methods cause artificial redistribution of the different chemical fractions (Anawar et al., 2010; Baeyens et al., 2003; Huang et al., 2015). As a result, the definition of bio available fraction is no longer possible.
  • Line 454. Remove "are mostly biophilic…. which"
  • Line 468. Check the whole section. The authors should avoid expressions like "data on the Cs-137 activity are listed in Table 3" or " Table 5 shows a proportion between the mobile and inert fractions of the radioactive and stable Cs in bottom sediments". They must indicate explicitly their results, showing the plot or table in brackets.
  • 480-481. Put the information of Tables 4 and 5 together. Rename fraction 1 and 2 as mobile and inert fraction, respectively.
  • 487-489. The use of background concentrations from local unpolluted sediments could be more appropriate than those from the earth's crust (Abrahim and Parker, 2008; de Souza Machado et al., 2018). Is there any information in the cited studies of Table 6? For instance, I notice that the work of Budko et al. (2017) contains values for some of the elements studied here. Make a comparison where possible.
  • Line 492. The information of Fig. 5 is hard to see. Remove or move to supplementary material.
  • Line 515. Be aware the typo: "Fig. 6c, d". Same as Fig 5, it is hard to see.
  • Line 537. The number of sampling sites in Fig. 7 is lower than in Figures 5 and 6 (12 vs 19). Why? Be consistent.
  • Lines 553-555. There is no point in displaying the graphics of ER and RI if the category of elements does not change. Replace these indicators with EF and Igeo and put all information within the same Figure (a-EF; b-Igeo; c-CF; d-DC). Maybe, it is not essential to include all elements for EF and Igeo Move any discarded information to supplementary material if necessary.
  • Lines 557-558. The sentence is a bit confusing. Re-phrase as follows:

"The examined sediments contain(ed) low levels of organic and inorganic carbon (Table 4), as is typical for the Arctic regions". Indicate some references. Two general aspects:

1) I personally use past and present simple for Results and Discussion section, respectively. But it is optional. Anyway, be consistent.

2) Try to avoid expressions like "according to our data" to simply provide figures. It is obvious. Such expression should be used for comparing and contrasting ideas, results, conclusions, etc.

  • Line 565-566. This is not true (see my comment above).
  • Line 569. Remove "authigenic". The term refers to mineral phases that are formed within sediment through diagenesis. Detrital Fe-Mn oxyhydroxides may also occur in the marine environment and are undistinguishable from the former by using extraction procedures. Most importantly, the authors missed sulfides as an important pool of this fraction. Monosulfides are completely solubilized with hot 6N HCl whereas other iron sulfides like greigite (F3S4) can be partially released to the solution. In contrast, Pyrite (FeS2) is not attacked and remains fairly unaltered during extraction (Cornwell and Morse, 1987; Rickard and Morse, 2005). Be aware some elements bound to organic matter and more crystalline oxyhydroxides may not fully extracted either (Chao, 1984; Hirner, 1992; Sutherland and Tack, 2003). This is important for further interpretations of the results (see my comments below).
  • Lines 581-583. Dissolved Fe2+ and hydrogen sulfide (H2S) are also involved in the reduction of Mn (IV) in anoxic sediments (Burdige, 1993; Myers and Nealson, 1988).
  • Lines 583-586. The direct comparison of results could be inexact. Bear in mind that the extraction procedures of the cited works are different from the one used here. In this study, the HCl-extractable fraction might contain Mn bound to carbonates and other reactive forms like organic matter or sulfides apart from Mn-bearing oxyhydroxides. An indirect way of getting some information about Mn-carbonates is seeking some kind of relationship between Sr/Ca and Mn/Al ratios (expressed in total contents). The Sr/Ca ratio is a proxy of biogenic carbonates which sometimes is related to increases in Mn/Al ratios in sediments (Jimenez-Arias et al., 2016). Explore this possibility.
  • Lines 587-597. It is not clear when the authors refer to their own results or are discussing data from other studies. Check. On the other hand, it is worth mention again that the extraction with hot 6N HCl does not extract sulfides totally (see above comment). This likely explain the observed differences between Fe and Mn. Pyrite (FeS2) is one of the main sinks of Fe in marine sediments (Berner, 1984) and considered one of their reactive forms (Poulton and Canfield, 2005; Raiswell, 2006). However, the extraction scheme of this study is not able to catch this feature and categorizes erroneously this pool as inert.
  • Lines 601-602. Unfortunately, a large part of oxyhydroxides were likely formed during drying of the samples. In addition, the drying causes a shift from amorphous to more crystalline forms (by aging process), increasing the alteration of natural conditions (Anawar et al., 2010; Baeyens et al., 2003; Huang et al., 2015; Larner et al., 2008). This provokes a significant impact on the chemical partitioning of many elements, preventing from drawing reliable conclusions. Moreover, the PCA analysis should be done again in case of fractions 1 and 2 would include.
  • Lines 603-617. As with Mn and Fe, the observed differences for Pb may arise from the utilization of different extraction procedures on each study. Nevertheless, the inclusion of a matrix correlation between the elements analysed can provide extra and useful information for this work.
  • Lines 618-620. The total concentrations for Bi are close to DL in most cases (Table S1) and, consequently, this finding does not deserve attention. Use this criterion for all elements.
  • Lines 627-636. Check grammar and coherence.
  • Lines 67-638. Again, I do not fully understand what they meant. Perhaps: "As with Cu, the distribution of Zn and Cd in the studied sediments are also controlled by organic matter". Check.
  • Lines 643-645. It is hard to believe this explanation considering the low levels of organic C reported in these sediments (Table 4). Even though it were a potential source (Sattarova et al., 2021), the amount of Cd that is accumulated in the primary producers would be minimal. More likely, Cd has its origin in some of the anthropogenic activities that take place in the surroundings as typically found in similar environments (Jafarabadi et al., 2017; Peña-Icart et al., 2017; Sharifinia et al., 2018; Sutherland, 2000).
  • Lines 645-651. As is by far one of the most affected elements by the aforementioned drying process according to the literature (Anawar et al., 2010; Huang et al., 2015; Larner et al., 2008). Thus, any interpretation involving oxyhydroxides must be precluded.
  • Lines 657-666. The conclusion is not based on their own results. Check grammar and coherence as well.
  • Lines 667-685. The conclusions are exclusively based on other works because results of this study are not provided. Remove.
  • Lines 686-687: Note some reactive forms are included in this inert fraction (see my comments above).
  • Lines 690-692.They cannot conclude with this because the extraction procedure is not selective for oxyhydroxides as discussed at length. Overall, this section is very speculative. The majority of conclusions are based on the results of other studies as pointed many times.
  • Line 693. This section need to be re-structured. First, total contents of heavy metals, results of PCA analysis and EF were given much more weight than other criteria for contamination such as CF, DC, ER or RI index. On the other hand, the authors pay too much attention to the comparison with earlier studies. In contrast, detailed discussions about the similarities and differences among the four areas is absent.
  • Lines 696-698. Remove this sentence (see one of my comments above concerning general aspects)
  • Lines 698-700. Re-phrase as follows (for similar reasons): "The concentrations of heavy metals in the Barents Sea sediments are lower than those reported in previous studies"
  • Lines 700-702. Indicate if there is some correlation between heavy metals and Al contents in order to support this statement. If so, link this sentence with the next one.
  • Lines 704-706: Move Table 6 to Suppl. material.
  • Line 784. Remove Fig. 8. The text is enough.
  • Lines 798. The whole section need to be shorten and re-think. I have the impression that the factors determining the patterns of distribution are unclear. The authors must explain the absence of correlation between Cs137 contents and grain size. Maybe, the organic matter contents might explain better their results. This kind of relationships are typically found in other works (Ligero et al., 2005, 2001).
  • Lines 799-802. Remove the expression "As it is known". It is a weird way to start a section. Check grammar and coherence as well.
  • Lines 802-804. Remove or move this sentence to Introduction.
  • Lines 805-822. The whole paragraph is a long description of the main findings reported by other researchers. Remove or link with their own results.
  • Lines 855-857. Remove Fig.9. The text is enough.
  • Lines 867-868. Such statement says nothing if the geochemical characteristics are not clearly identified.
  • Line 870. Geochemical fractions of the radionuclides suffer from the same failures than total elements contents. Remove the whole section. Nevertheless, total contents of Pu-239 could be used in the previous section.
  • Line 907. It is too long. Shorten.

 References cited:

Abrahim, G.M.S., Parker, R.J., 2008. Assessment of heavy metal enrichment factors and the degree of contamination in marine sediments from Tamaki Estuary, Auckland, New Zealand. Environ. Monit. Assess. 136, 227–38. https://doi.org/10.1007/s10661-007-9678-2

Anawar, H.M., Mihaljevic, M., Garcia-Sanchez, A., Akai, J., Moyano, A., 2010. Investigation of sequential chemical extraction of arsenic from sediments: Variations in sample treatment and extractant. Soil Sediment Contam. 19, 133–141. https://doi.org/10.1080/15320380903548466

Baeyens, W., Monteny, F., Leermakers, M., Bouillon, S., 2003. Evaluation of sequential extractions on dry and wet sediments. Anal. Bioanal. Chem. 376, 890–901. https://doi.org/10.1007/s00216-003-2005-z

Berner, R.A., 1984. Sedimentary pyrite formation : An update. Geochim. Cosmochim. Acta 48, 605–615. https://doi.org/10.1016/0016-7037(84)90089-9

Budko, D.F., Demina, L.L., Lisitzin, A.P., Kravchishina, M.D., Politova, N. V., 2017. Occurrence forms of trace metals in recent bottom sediments from the White and Barents Seas. Dokl. Earth Sci. 474, 552–556. https://doi.org/10.1134/S1028334X17050014

Burdige, D.J., 1993. The biogeochemistry of manganese and iron reduction in marine sediments. Earth-Science Rev. 35, 249–284. https://doi.org/10.1016/0012-8252(93)90040-E

Chao, T.T., 1984. Use of partial dissolution techniques in geochemical exploration. J. Geochemical Explor. 20, 101–135.

Choi, K.Y., Kim, S.H., Chon, H.T., 2012. Relationship between total concentration and dilute HCl extraction of heavy metals in sediments of harbors and coastal areas in Korea. Environ. Geochem. Health 34, 243–250. https://doi.org/10.1007/s10653-011-9425-z

Cornwell, J.C., Morse, J.W., 1987. The characterization of iron sulfide minerals in anoxic marine sediments. Mar. Chem. 22, 193–206.

de Souza Machado, A.A., Spencer, K.L., Zarfl, C., O’Shea, F.T., 2018. Unravelling metal mobility under complex contaminant signatures. Sci. Total Environ. 622–623, 373–384. https://doi.org/10.1016/j.scitotenv.2017.11.239

Hirner, A. V., 1992. Trace element speciation in soils and sediments using sequential chemical extraction methods. Int. J. Environ. Anal. Chem. 46, 77–85. https://doi.org/10.1080/03067319208026999

Huang, G., Chen, Z., Sun, J., Liu, F., Wang, J., Zhang, Y., 2015. Effect of sample pretreatment on the fractionation of arsenic in anoxic soils. Environ. Sci. Pollut. Res. 22, 8367–8374. https://doi.org/10.1007/s11356-014-3958-5

Jafarabadi, A.R., Bakhtiyari, A.R., Toosi, A.S., Jadot, C., 2017. Spatial distribution, ecological and health risk assesment of heavy metals in marine surface sediments and coastal seawaters of fringing coral reefs of the Persian Gulf, Iran. Chemosphere 185, 1090–1111.

Jimenez-Arias, J.L., Mata, M.P., Corzo, A., Poulton, S.W., Ma rz, C., Sanchez-Bellon, A., Martinez-Lopez, J., Casas-Ruiz, M., Garcia-Robledo, E., Bohorquez, J., Papaspyrou, S., 2016. A multiproxy study distinguishes environmental change from diagenetic alteration in the recent sedimentary record of the inner Cadiz Bay (SW Spain). The Holocene 26, 1355–1370. https://doi.org/10.1177/0959683616640046

Larner, B.L., Palmer, A.S., Seen, A.J., Townsend, A.T., 2008. A comparison of an optimised sequential extraction procedure and dilute acid leaching of elements in anoxic sediments , including the effects of oxidation on sediment metal partitioning. Anal. Chim. Acta 8, 147–157. https://doi.org/10.1016/j.aca.2007.12.016

Ligero, R.A., Barrera, M., Casas-Ruiz, M., 2005. Levels of 137Cs in muddy sediments on the seabed in the Bay of Cadiz (Spain). Part II. Model of vertical migration of (137)Cs. J. Environ. Radioact. 80, 87–103. https://doi.org/10.1016/j.jenvrad.2004.06.006

Ligero, R.A., Ramos-Lerate, I., Barrera, M., Casas-Ruiz, M., 2001. Relationships between sea-bed radionuclide activities and some sedimentological variables. J. Environ. Radioact. 57, 7–19.

Mclaughlin, M.J., Zarcinas, B.A., Cook, N., 2000. Communications in Soil Science and Plant Analysis Soil testing for heavy metals 37–41.

Myers, C.R., Nealson, K.H., 1988. Microbial reduction of manganese oxides: Interactions with iron and sulfur. Geochim. Cosmochim. Acta 52, 2727–2732. https://doi.org/10.1016/0016-7037(88)90041-5

Peña-Icart, M., Pereira-Filho, E.R., Fialho, L.L., Nóbrega, J.A., Carlos, A.-H., Bolaños-Alvarez, Y., Pomares-Alsonso, M.S., 2017. Combining contamination indexes, sediment quality guidelines and multivariate data analysis for metal pollution assesment in marine sediments of Cienfuegos Bay, Cuba. Chemosphere 168, 1267–1276. https://doi.org/10.1016/j.chemosphere.2016.10.053

Poulton, S.W., Canfield, D.E., 2005. Development of a sequential extraction procedure for iron: implications for iron partitioning in continentally derived particulates. Chem. Geol. 214, 209–221. https://doi.org/10.1016/j.chemgeo.2004.09.003

Raiswell, R., 2006. Towards a global highly reactive iron cycle. J. Geochemical Explor. 88, 436–439. https://doi.org/10.1016/j.gexplo.2005.08.098

Rickard, D., Morse, J.W., 2005. Acid volatile sulfide (AVS), Marine Chemistry. https://doi.org/10.1016/j.marchem.2005.08.004

Sattarova, V., Aksentov, K., Astakhov, A., Shi, X., Hu, L., Alatortsev, A., Mariash, A., Yaroshchuk, E., 2021. Trace metals in surface sediments from the Laptev and East Siberian Seas: Levels, enrichment, contamination assessment, and sources. Mar. Pollut. Bull. 173, 112997. https://doi.org/10.1016/j.marpolbul.2021.112997

Sharifinia, M., Taherizadeh, M., Namin, J.I., Kamrani, E., 2018. Ecological risk assesment of trace metals in the surface sediments of the Persian Gulf and Gulf of Oman: Evidence from subtropical estuaries of the Iranian coastal waters. Chemosphere 191, 485–493.

Snape, I., Scouller, R.C., Stark, S.C., Stark, J., Riddle, M.J., Gore, D.B., 2004. Characterisation of the dilute HCl extraction method for the identification of metal contamination in Antarctic marine sediments. Chemosphere 57, 491–504. https://doi.org/10.1016/j.chemosphere.2004.05.042

Sutherland, R.A., 2000. Bed sediment-associated trace metals in an urban stream, Oahu, Hawaii. Environ. Geol. 39, 611–627. https://doi.org/10.1007/s002540050473

Sutherland, R.A., Tack, F.M.G., 2003. Fractionation of Cu, Pb and Zn in certified reference soils SRM 2710 and SRM 2711 using the optimized BCR sequential extraction procedure. Adv. Environ. Res. 8, 37–50. https://doi.org/10.1016/S1093-0191(02)00144-2

Reviewer 2 Report

The authors produce a thorough analysis of recent sampling data from three arctic sea areas. The data collection methodology, analysis by PCA and comparison to geo-accumulation index and enrichment factors seem sound and data support the conclusions presented in this paper. The paper is, however, rather wordy and occasionally boarders on repetitive, the discussion and conclusions could perhaps be more succinct. Minor comments are included below. 

  1. The abstract, for example is long and reads more like an introduction. eg lines 15-18 “In this study, we aimed to quantify the principal geochemical frac-15 tions of a large number of elements (including the heavy metals, and Cs-137 and Pu-239 techngenic 16 radionuclides) in bottom sediments, as well as to estimate the current contamination of the surface 17 bottom sediments from some areas of the Barents, Kara, Laptev, and East-Siberian Seas.” Really belong in the introduction. An abstract should be a brief summary of the key findings not an explanation of the aims and intentions of the study.
  2. The word ‘Forms’ is used repeatedly and in this context and as a chemist I take it to mean ‘speciation’ Definition: Chemical speciation refers to the distribution of an element amongst chemical species in a system. It is critical for understanding chemical toxicity, bioavailability, and environmental fate and transport.

This might be worth either changing forms to speciation or defining the use of the word ‘form’ where first used (I think line 78 and 99).

Line 79: do you mean speciation rather than specification

  1. The methodology. Under the heading “The radionuclide analysis” there is only a description of how Cs-137 was measured and no mention of Pu. Later the Pu measurement procedure was described after description of a modified sequential extraction procedure. This took several reads to follow what had been done. I suggest that the method for measuring the activity of Pu is moved to the section above with Cs-137.
  2. Line 254 (Grounded)
  3. I was a little confused by the ‘total’ line on the graphs in figure 3. This is clearly not mobile plus inert fraction but if not then how is it plotted?
  4. The terms ‘reactive’, ‘mobile’ and ‘fraction1’ are used interchangeable to describe that portion of the elements that is released by 6M acid addition. Perhaps just use mobile to be clear and certainly get rid of ‘fraction 1’ used in table 4 and 5 as this is unnecessary (just say mobile and inert). Later the authors refer to “reactive forms” when they have been talking about “mobile fraction”. Perhaps better to pick one term.
  5. Similarly there is mention of subgroups and then groups. Subgroups are then not mentioned again. The mixed terminology is confusing.
  6. Figure 6. Rather than write EC, H-EX etc and then have a key it might just be easier to write the words on the graph.

Reviewer 3 Report

This manuscript made a through investigation on the contents of metals of the sediments in the sediments of the Barents, Kara, Laptev and East-Siberian Seas. The data  could be an important basic information. However, the scientific value is kind of lack, and the novelty needs to be more clearly described. Despite the PCA analysis, maybe the authors could try to dig more valuble information from the obtained data, such as their geographic distribution and their geographic correlation, which may help disclose the relationships among the elements and the human activities. 

Further, as said in the introduction, it might be interesting to know the  anthropogenic pressure on the coastal environment, however, there was no quantitative description of the pressure.

Round 2

Reviewer 1 Report

I am very glad that my comments and suggestions have been useful and taken in good faith. This new version is certainly enhanced with the made changes. However, the whole manuscript needs to be edited for expression and grammar, especially the Discussion section. This and other considerations are detailed below:

  • Lines 51-53. Relocate the information of this sentence (including reference number 5) to the two previous sentences and remove it.
  • Lines 54-55. Try to relate this point with the previous sentence. For instance, re-phrase as follows (or something like that): "Due to the great danger of technogenic radionucleides (in particular Cs-137) to natural ecosystems, as consequence their ecotoxicity, the acknowledgement of their distribution features and speciation in Arctic sediments is an urgent modern task".
  • Connect the fragment of "data on the content of heavy metals and radioactive nuclides in the sediments of the Eurasian Arctic shelf seas remain not completely investigated to date (Lines 59-61)" to the previous paragraph using linkers: However, Despite of that, Nonetheless, etc. Remove the rest (Lines 56-59).
  • Line 337. The Pearson correlation coefficient must be denoted in lower case (r). Regardless of whether the relationship is negative or positive, the range of 0.3-0.6 indicates medium strength. Thus, replace "insignificant" with "moderate”.
  • Lines 340-347. The differentiation between Fig. 3a and 3b is unnecessary now.
  • Line 403. Be aware the typo: “RI”.
  • Line 417. Be aware the typo: “RI”.
  • Line 434. It should be noticed that clays are already included in the mineralogical group of aluminosilicates. Replace "aluminosilicates" with "feldspars" to be precise.
  • Line 436. Replace "the low quantity" with "low quantities"
  • Lines 445-446. This sentence must go before (in Line 437)
  • Lines 446-447. The sentence needs checking. Among other things, the term "mineral complexes" could lead to misinterpretations. Re-phrase as follows: “Apparently, the main factor determining the geochemical characteristics of Arctic sediments results from the composition and textural features of minerals".
  • Lines 447-450. Actually, it tells you that most of these elements are prevalent in the terrigenous fraction. Re-phrase
  • Lines 450-452. Give an explanation. In my opinion, the association could be an indication of the prevalence of heavy minerals in the studied sediments (Achab and Gutierrez-Mas, 2009).
  • Lines 462-464. Check the meaning of this sentence. I guess the authors meant: "Despite the Cu distribution in the marine sedimentation is commonly controlled by its association with organic matter [49–51], Cu is found in this study among the elements bound to Fe-Mn oxyhydroxides".
  • Lines 464-465. Check grammar and coherence. For instance, the sentence could be written as follows: "Similar results were found in the study of…" (or something like that). Note the Pearson correlation coefficient must be denoted as r.
  • Lines 462-471. Remove the two last sentences of this paragraph: "On the other hand, provided ... oxidants of organic matter". This weakens the case for Fe-Mn oxyhydroxides.
  • Lines 472-474. Cd and As show correlation with other elements, albeit moderate (see Table S3). Remove this sentence.
  • Lines 475-476. This is not true (see my previous comment).
  • Lines 476-477. R instead of r, again. Please, check this error throughout the text. Figure 3 or Table S3? Clarify. Note a symbol has been erroneously included as well.
  • Lines 483-484. Remove from the same reasons
  • Line 485. Replace "a presence" with "its occurrence"
  • Lines 487-488. I do not fully understand what they meant. Anyway, the reference number 58 does not support the argument either.
  • Lines 488-489. None of these arguments are totally convincing. In my opinion, the inclusion of As in a separate geochemical association is very likely a consequence of its anthropogenic origin (see next section: lines 574-576).
  • Line 496. Replace "a formation of the" with "the formation of"
  • Lines 503-511. Could these findings be interpreted as a reduction of contamination in the last years? If so, point it out on the text.
  • Line 536. Be aware the typo: “RI”.
  • Lines 548-550. Check grammar and coherence.
  • Lines 567-592. Basically, the main point is that contamination by As comes from other anthropogenic sources rather than military tests. Re-phrase and shorten the whole paragraph.
  • Line 592. Include a small conclusion as in section 4.1.
  • Line 602. Replace the expression of "From our data, it can be seen that" with "In our study,". Note I left a remark in the previous review about it. Check similar expressions throughout the manuscript.
  • Line 617. Be aware the typo: "Despite the fact that,"
  • Lines 617-620. Include references to support the argument.
  • Lines 621-626. Replace the whole paragraph with the following (and missing) piece of text: "In general, a correlation between Cs-137 and grain size composition can be noticed. In those samples where the lowest contents of the pelite fraction were found, the content of Cs-137 was minimal. The rest of the sediment samples contained more than 75% wt. pelite. Probably, variations in content of Cs-137 were attributed to a presence of potential local sources of pollution, rather than the content of a fine-grained fraction. In the literature, a general tendency for cesium affinity for clay minerals was described, but there is no clear direct correlation with grain size [Kuzmenkova et al., 2019]. Relationship between content of Cs-137 and total organic carbon (TOC) was not found for our samples, due to the lack of statistical data on TOC. In addition, the examined Arctic sediments have very low TOC content. A similar relationship could be traced to soils with typical high TOC content”.
  • Line 632. A small conclusion is needed as well. Nevertheless, my central objection of this section is that the ideas and arguments are not well linked with each other. Please, check thoroughly.
  • Line 642. Replace "By our data," with "In this work,"
  • Lines 643-646. Remove "Presumably" and "of Cd". Be aware that the aforementioned origin for Cd is not certain (or at least it is not the only one). Moreover, the As distribution in these sediments has more to do with anthropogenic sources than its chemical features. Check.
  • Line 650. A couple of lines below it is stated that the As enrichments are caused by anthropogenic sources. Be consistent.
  • Lines 648-652. Note specific cases of diagenetic distribution in the studied elements are absent in the discussion, or at least, they were not mentioned with such words. Check.
  • Lines 664-672. Use this paragraph as a guide to the final conclusion of section 4.3 (see my comment above).

Reference cited

Achab, M., Gutierrez-Mas, J.M., 2009. Heavy minerals in modern sediments of the bay of Cadiz and the adjacent continental shelf (southwestern Spain): nature and origin. Thalassas 25, 27–40.

Author Response

Dear Reviewer,

We are very glad that you have appreciated our revision of the manuscript. Certainly, your additional recommendations made it possible to improve our manuscript. All comments have been responded to. We have also edited the expression and grammar of the manuscript. Please see the attachment file.
